# From Electron Imbalance to Network Collapse: Decoding the Redox Code of Ischemic Stroke for Biomarker-Guided Precision Neuroprotection

**DOI:** 10.3390/ijms262210835

**Published:** 2025-11-07

**Authors:** Ionut Bogdan Diaconescu, Adrian Vasile Dumitru, Calin Petru Tataru, Corneliu Toader, Matei Șerban, Răzvan-Adrian Covache-Busuioc, Lucian Eva

**Affiliations:** 1Department of Anatomy, “Carol Davila” University of Medicine and Pharmacy, 050474 Bucharest, Romania; 2Department of Pathology, Faculty of Medicine, “Carol Davila” University of Medicine and Pharmacy, 030167 Bucharest, Romania; 3Department of Opthamology, “Carol Davila” University of Medicine and Pharmacy, 020021 Bucharest, Romania; 4Central Military Emergency Hospital “Dr. Carol Davila”, 010825 Bucharest, Romania; 5Puls Med Association, 051885 Bucharest, Romania; 6Department of Neurosurgery, “Carol Davila” University of Medicine and Pharmacy, 020021 Bucharest, Romania; 7Department of Vascular Neurosurgery, National Institute of Neurology and Neurovascular Diseases, 077160 Bucharest, Romania; 8”Nicolae Oblu” Clinical Hospital, 700309 Iasi, Romania

**Keywords:** ischemic stroke, redox code, mitochondrial reprogramming, ferroptosis, reverse electron transport, microvascular no-reflow, NETosis, adaptive redox-guided therapy, biomarkers, translational neuroprotection

## Abstract

Ischemic stroke remains one of the most catastrophic diseases in neurology, in which, due to a disturbance in the cerebral blood flow, the brain is acutely deprived of its oxygen and glucose oligomer, which in turn rapidly leads to energetic collapse and progressive cellular death. There is now increasing evidence that this type of stroke is not simply a type of ‘oxidative stress’ but rather a programmable loss-of-redox homeostasis, within which electron flow and the balance of oxidants/reductants are cumulatively displaced at the level of the single molecule and at the level of the cellular area. The advances being made in cryo-electron microscopy, lipidomics, and spatial omics are coupled with the introduction of a redox code produced by the interaction of the couples NADH/NAD^+^, NADPH/NADP^+^, GSH/GSSG, BH_4_/BH_2_, and NO/SNO, which determine the end results of the fates of the neurons, glia, endothelium, and pericytes. Within the mitochondria, pathophysiological events, including reverse electron transport, succinate overflow, and permeability transition, are found to be the first events after reperfusion, while signals intercommunicating via ER–mitochondria contact, peroxisomes, and nanotunnels control injury propagation. At the level of the tissue, events such as the constriction of the pericytes, the degradation of the glycocalyx, and the formation of neutrophil extracellular traps underlie microvascular failure (at least), despite the effective recanalization of the vessels. Systemic influences such as microbiome products, oxidized lipids, and free mitochondrial DNA in cells determine the redox imbalance, but this generally occurs outside the brain. We aim to synthesize how the progressive stages of ischemic injury evolve from the cessation of flow to the collapse of the cell structure. Within seconds of injury, there is reverse electron transport (RET) through mitochondrial complex I, with bursts of superoxide (O_2_•^−^) and hydrogen peroxide (H_2_O_2_) being produced, which depletes the stores of superoxide dismutase, catalase, and glutathione peroxidase. Accumulated succinate and iron-induced lipid peroxidation trigger ferroptosis, while xanthine oxidase and NOX2/NOX4, as well as uncoupled eNOS/nNOS, lead to oxidative and nitrosative stress. These cascades compromise the function of neuronal mitochondria, the glial antioxidant capacity, and endothelial–pericyte integrity, leading to the degradation of the glycocalyx with microvascular constriction. Stroke, therefore, represents a continuum of redox disequilibrium, a coordinated biochemical failure linking the mitochondrial metabolism with membrane integrity and vascular homeostasis.

## 1. Prologue: Redox Reprogramming as the Missing Dimension in Acute Ischemic Stroke

Approximately 12 million people suffer their first stroke each year, and 6.5 million die from stroke; it is now the second leading cause of death and the largest single cause of persistent long-term neurological disability. Ischemic stroke accounts for approximately 85% of all stroke cases and is caused by an acute loss of blood flow to the brain through one of three common mechanisms: (a) large artery atherosclerosis, (b) cardio-embolic pathways, and (c) small-vessel occlusive disease [1,2]. The major modifiable risk factors for stroke are hypertension, diabetes mellitus, atrial fibrillation, dyslipidemia, obesity, and smoking; however, the contribution of non-modifiable risk factors, including age, sex, and genetic predisposition causing increased vulnerability, should also be considered. Rapid assessment of stroke severity can be achieved with standardized measures, including both the NIHSS (National Institutes of Health Stroke Scale) and mRS (modified Rankin Scale) scores; the NIHSS assesses neurological impairment at the time of evaluation, while the mRS assesses the long-term functional outcome [3]. Despite advances in the acute management and vascular recanalization of strokes, the heterogeneity in stroke outcomes underscores that the ultimate neurological deficit experienced by individuals who have had a stroke cannot be solely attributed to macrovascular events [4]. The microvascular events that occur immediately after stroke onset, including energy failure, oxidative imbalance, and neurovascular dysfunction, represent the true determinants of recovery or collapse after a stroke. Therefore, understanding these microscopic processes is critical to translating successful vascular interventions into improved neuronal survival [5].

Acute ischemic stroke remains among the most severe disorders of the central nervous system, where minutes are a lifetime. Over the past thirty years, while we have witnessed real advances in acute management of strokes, with intravenous thrombolysis and mechanical thrombectomy improving the prognosis for large-vessel occlusions, the original paradox remains: reopening the artery does not equal restoring the brain. Clinical reality has time and again taught us that technically successful reperfusion is often followed by infarction growth, malignant edemas, or persistent functional neurological deficit. The images of contrast passing through reopened vessels hide persistent hypoxia, neuronal death, and network failure at the tissue level [6]. Extensive experimental investigation over the last three decades has provided a clearer understanding of the mechanisms by which reperfusion might lead to subsequent cellular necrosis. The rapid reestablishment of the delivery of both oxygen and substrates causes the elaboration of an explosive amount of reactive oxygen and nitrogen species (ROS/RNS) from mitochondrial, xanthine oxidase, and NADPH oxidase sources [7]. Mitochondrial complex I dysfunction, with succinate accumulation and reverse electron transport (RET), has been postulated as the major source of reperfusion-related superoxide. In parallel with these events, glutamatergic excitotoxicity and intracellular calcium overload activate other enzymatic sources of ROS, including nitric oxide synthases, and lipoxygenases, enhancing lipid peroxidation and the loss of mitochondrial membrane integrity [8]. Endogenous antioxidant systems—superoxide dismutase, catalase, glutathione peroxidase and thioredoxin—become rapidly saturated with their substrates culminating in oxidative damage to proteins and DNA, structural disruption of membrane integrity and an eventual programmed neuronal death. Collectively, these interrelated systems have established oxidative stress as a central, yet still incompletely defined, concept in the biology of ischemic brain injury [9]. This paradox compels us to investigate further, into mechanisms hidden behind angiography and cannot be illuminated by gross flow alone. For decades, the most obvious explanation for this hidden pathology was oxidative stress, an umbrella term incorporating the uncontrolled generation of ROS and RNS. That, indeed, reperfusion “drowns” neurons in toxic oxidants captured the imagination and launched whole therapeutic development efforts. But even after many unsuccessful antioxidant trials, we now recognize that, while attractive, this concept was ultimately incomplete [10]. This review seeks to supplant that outdated idea with a more precise, and what we think is more productive, idea: that stroke is not governed by random oxidative chaos but by a programmable redox trajectory, a time-based chain of molecular events that dictate whether reperfusion yields injury or recovery.

The redox trajectory begins with the mitochondria, which will leave a biochemical footprint of trapped metabolites after ischemia. Specifically, succinate, a Krebs cycle intermediate, immediately upon anaerobic metabolism spikes in concentration. When succinate causes reperfusion, it acts as the match that lights off the hyperacute redox detonation [11]. This is happening because succinate catalyzes RET at complex I of the respiratory chain, forcing electron flow to move backward to the flavin mononucleotide site and causing the electron leak to superoxide. In 2024, utilizing calibrated reperfusion models, it was shown that reverse electron transport (RET) at the level of complex I regulates some 60–70% of the measured ROS burst, which is seen in the initial period of the first seconds of reperfusion [9,12]. This has been shown using quantitative methodologies utilizing measurements of mitochondrial oxygen consumption, residual succinate and measurements of ROS flux, which showed that one major source of oxidants from reperfusion is RET but not the only source. The ischemic phase is also associated with destabilization in complex I due to increased unbinding of the FMN, which leads to increased electron leak and decreased catalytic turnover, which will lead to increased extreme oxidative stress should oxygen then be reintroduced to the mitochondria [13]. Therefore, RET is not an abstract descriptor for oxidative stress but a discrete micro-site and in vitro validated phenomenon. Though mitochondrial in origin, its significance is greater than the mitochondria. Transporters of succinate found in astrocytes, neurons and endothelium traffic succinate between compartments, generating intercell redox gradients. When viewed through this lens, RET is not a mishap of mitochondrial chemistry but a metabolic mishandling of a systemic metabolite that contextualizes the fate of the tissue [14]. Oxidants generated in RET build up on cell membranes, activating ferroptosis cell death, a distinct form of necrosis regulated by iron-dependent lipid peroxidation. Apoptosis was previously considered to be the canonical cell death path in stroke; ferroptosis is now in the limelight. Several high-impact original studies presented in 2024 clearly show that ferroptosis inhibitors, Ferrostatin-1 and Liproxstatin-1, reduced infarct volumes, protected neurons, and oligodendrocytes, and decreased oxidized phospholipids [15,16]. Ferroptosis is uniquely time-specific: it only arises hours post-reperfusion, when apoptosis takes days to decades, or in a proximal timeframe to the hyperacute redox storm. Also, unlike apoptosis, ferroptosis may be spatially sparse in the brain, torturing oligodendrocytes and myelinated axons with more select vulnerability, and it correlates with the gross distortion of white matter that is observed locally and clinically [17]. In addition, unique mechanisms provide even more support: lipoxygenases such as ALOX15 and ALOX12 can stimulate lipid per-oxidation; astrocytic ferroptosis is associated with reduced glutamate uptake function and potential excitotoxicity; crosstalk of ferroptosis and necroptosis in an iron-rich environment implicates iron, a dual activator of cell death modality [18]. The voltage-dependent anion channel (VDAC) of mitochondria may relate to ion flux coupled with the fatty acid profile tethered to susceptibility. Looking back, these singular findings breathe life into ferroptosis not as another modality but, rather, the redox-based harbinger of ischemic cell death [19].

As molecular injury progresses, intercellular redox imbalance is no longer limited to neurons but spreads to the entire neurovascular unit. The accumulation of iron-dependent lipid peroxides and the depletion of glutathione appear to be important mediators of ferroptosis, a form of regulated necrosis, which links the oxidative stress of the mitochondria to the subsequent destruction of the tissue. Several studies in the models of ischemia reperfusion have shown that the application of inhibitors such as Ferrostatin-1 and Liproxstatin-1 significantly reduces the volume of the infarct, diminishes the extension of lipid peroxidation and protects the neurons and oligodendroglia [20,21,22]. Ferroptosis is, thus, a terminal node of redox imbalance: it occurs at a time when the neutral mechanisms fail and the flux of electrons into the mitochondria is disordered and linked to the biochemical generation of (might one say free) radical species to the initiation of structural damage to the tissues of the brain. Following the cascade of ferroptotic and oxidative injury, the microcirculation in turn becomes a secondary victim [23]. It has been several years since the no-reflow phenomenon was established: macrovascular recanalization does not necessarily ensure the re-establishment of capillary flow. There are two predominant determinants. One is provided by the pericytes, which are the mural cells within the capillary region controlling in the main capillary diameter [24]. In consequence of ischemia, these cells suffer mitochondrial depolarization and intracellular overload with calcium, causing persistent constriction of the lumen, even when the upstream recanalization has been completed [25]. The experimental application of inhibitors of mTOR, with rapamycin, has been shown to be able to relax the pericyte elements and re-establish perfusion of the microcirculation, indicating that this is potentially a reversible element in microvascular inability to function correctly [26]. The second factor has to do with the endothelial glycocalyx, a fine and delicate layer of carbohydrate elements, which will be rapidly destroyed during the period of reperfusion. The disintegration of this fine structure produces an alteration in the sensorium for shear, increases the adhesion of leucocytes and causes an alteration in the permeability of the vessel, the markers of which are syndecan-1 and heparan sulphate, which can be measured as markers of plasma injury to the glycocalyx within the circulation [27].

The loss of this basis of the glycocalyx will also have the effect of interfering with the endothelial mechanosensor elements such as Piezo1 and TRPV4 and will cause further alterations in the calcium signal, which will result in greater oxidative stress within the endothelium [28]. Here, at this moment, redox imbalance will convert the microcirculation into a battlefield, in which energy failure, iron-catalyzed lipid oxidation and nitric oxide dysregulation will rage. Under conditions of the physiological redox tone, nitric oxide produced from endothelial nitric oxide synthase (eNOS) will ensure that vasodilation, inhibition of the platelets and maintenance of the barrier function are retained. In the event of the oxidation of BH_4_ to BH_2_, a degree of uncoupling of eNOS will increase the product of eNOS, which will then be superoxide rather than nitric oxide, converting an anti-vasoprotective agent into a radical generator [29]. The improvement in the stability of the microcirculation can be ensured by re-establishing the BH_4_/BH_2_ ratio, as described in the endothelial experiments of late, which will re-couple the enzyme eNOS and restore microvascular stability.

Apart from these effects on eNOS, the early activation of neuronic nitric oxide synthase (nNOS) in the brain during early reperfusion will lead to an increase in the glutamate-induced excitotoxicity, while the delayed induction of the inducible nitric oxide synthase (iNOS) in glia will maintain a state of destructive nitrosative stress lasting for days [30,31]. These processes must be viewed as defining a continuum: the pathological state of mitochondrial redox failure moves towards ferroptotic vulnerability, followed in turn by microvascular injury, which will then further extend hypoxia, while the presence of an imbalance in the nitric will render decisions as to whether the tissue may heal itself or die. Viewed in this sequence, the redox state emerges as the central coordinating element from which will be derived both cellular and microvascular results.

Collectively, these studies provide evidence that ischemic stroke develops over several redox stages with unique pathophysiology and potential for intervention. Hyperacute (first minutes) is characterized by both RET at complex I and ferroptosis initiation; also seen during hyperacute are pericyte constriction, glycocalyx breakdown, and activation of nNOS. The acute–subacute (hours) phase is associated with adaptive gene expression driven by Nrf2, HIF-1α, and FoxO3a as attempts at mitochondrial repair, with the recoupling of eNOS and limited restoration of glycocalyx, to stabilize vasculature. Finally, the chronic (days–weeks) phase of ischemic stroke is characterized by continued activation of iNOS and, therefore, sustained nitrosative stress and limited potential for recovery. An understanding of the sequence of redox injury will allow for the creation of a temporal map of redox injury, allowing for stage-specific interventions (from early RET and ferroptosis inhibitors to later microvascular stabilizers and anti-nitrosative therapy) [32].

By the same token, we provide this introduction not to reach conclusions, nor to speculate, but we provide you with an easily readable summary. Our aim is to synthesize a profound conceptual innocence. Misrouted electrons and mitochondria, ferroptosis susceptibility, microvascular collapse, nitric oxide signaling here, thrombo-inflammatory leave of absence, and the mRNA transcription algorithm configure immersive as multi-scale. Therefore, our hopes are to produce and elicit thoughtful consideration of an invaluable model and rehabilitation of trial-based systems using crafty and discussion memory. Investigative rescue programs for clarity or what went wrong, as an oil sheen process like stroke was failure re-imagined not chaos oxidative stress, but theoretically on an observably programable redox-linked behavior, could inspire a more precise and sustainable rehabilitation trajectory.

## 2. The Redox Code of the Ischemic Brain

The ischemic stroke is not an end to perfusion but an abrupt reorganization of the energy-organizational circuitry of the brain. The denuding of oxygen and glucose is such that every element of the neurovascular unit (NVU) must have its electron economies reorganized, with some metabolic currencies collapsing and some going into deviation. Instead of the non-contributive title of oxidative stress, we solicit the redox code, a multi-limited syntax of the electron economies of NADH/NAD, NADPH/NADP, GSH/GSSG, BH_4_/BH_2_, NO/SNO, whose relative hemi-quantities and spatial organizations form the destinies of neurons, glia, endothelium and pericytes [33]. In Figure 1, we see the schematic branching of such codes themselves, incorporated as they are in the molecular, vascular, immune and systems levels. The synthesis below gives meanings to how any axis can co-operate on the ischemic damage or on recovery by interaction with other structural instruments and translational possibilities.

### 2.1. NADH/NAD^+^: Surplus Electrons and the Shifting Architecture of Complex I

Ischemia itself immediately alters the NADH/NAD ratio to be a reducing type, for example, to inhibit oxidative phosphorylation, enhancing glycolysis. The mitochondrial complex I attains the inactive D-state when the flavin mononucleotide (FMN) is dissociated from the whole element and the tunneling of electrons rendered unstable [34]. The reperfusion with the re-entry of oxygen into the mitochondria offers a reversion transport of electrons (RTE) to overt redox of the pool of ubiquinone, giving superoxide bursts at the FMN component [35].

Cryo-electron microscopy (EM) structural studies have shown other slip conformers of the complex I, which increase such instability [36]. The succinate derived from the excessive accumulation of aminobutyric acids, which themselves are spatially re-dispensed by the SLC13A5/SLC13A3 across astrocytes and the endothelium in the blood–brain barrier, makes for redox spatial differentials across the NVU. Because of this loss of additional NAD, not only is the bioenergetics impinged but also the activation of poly-ADP ribose polymerase (PARP) uses NAD for DNA repair, whilst inhibition of the sirtuins along altered histone acetylation causes a rewiring of the transcription from the bioenergetic elements to those associated with the apoptotic [37,38].

Hence, the NADH/NAD element links together (inextricable) the destructive metabolism with the epigenetic drift. NAD^+^ precursors (nicotinamide riboside, mononucleotide) restore these circuits, but imaging of NADH autofluorescence demonstrates the compartmentation of this rudimentary determination of cellular state, not universal sensing [39].

Hyperglycemia aggravates this inconsonance by depleting NADPH via the pentose phosphate pathway and/or by donating reducing equivalents to NOX rather than peroxidase detoxication [40,41]. In neurons, NADPH is rather directed to lipid hydroperoxides, diminishing the threshold towards ferroptosis death [42,43,44,45]. The collapse of the GSH/GSSG axis bespeaks the affair of crossing the antioxidant firewall. With GPX4 disabled, lipid hydroperoxides accumulate, leading to ferroptosis cell death [46]. The failure, moreover, of either of the four ferroptoses prevents GPX4–GSH, FSP1–CoQ10, GCH1–BH_4_, DHODH–CoQ, leading to runaway oxidative necrosis [46,47]. Membrane remodeling enzymes (ACSL4, LPCAT3) enrich polyunsaturated species, while ALOX12/15 and dysfunctional iPLA_2_β propagate peroxidation [43]. Ferroptosis occurs in neurons, astrocytes and oligodendrocytes, which interfere with glutamate detoxication and the stability of white matter. In lipidomic profiling in 2024, it was shown that oxidized phosphatidylethanolamines (e.g., 15-HpETE-PE) are traceable in plasma in reveneptosis [44]. Hence, the collapse of GSH/GSSG is not merely the loss of antioxidative defense but the total loss of redox conservation.

At the vascular front, the BH_4_/BH_2_ ratio determines the fidelity of nitric-oxide synthase (NOS) coupling. Proper BH_4_ provides the dedicand to the substrate for the generation of NO. Its oxidized form, BH_2_, uncouples NOS to then produce superoxide. Ischemia reduces GTP cyclohydrolase I activity and alters the availability of BH_4_, while oxidative stress increases the rate of oxidation of BH_4_ [45]. The isoform specificities promote deleterious sequele: nNOS uncoupling increases excitotoxicity, eNOS uncoupling loses endothelial integrity, while iNOS uncoupling leads to prolonged nitrosative stress. Clinically, the concentrations of BH_4_/BH_2_ in the plasma correlate with the degrees of stroke and microvascular dysfunction, while supplementation with either sapropterin or sepiapterin restores perfusion and decreases the infarct burden. Increased expression of arginase depletes substrate formulation while also increasing uncoupling, and NOS monomerization is a signal of structural failure [46,47]. Therefore, BH_4_/BH_2_ has the functional role of being gauges of fidelity of the vasculature, translating biochemical redox imbalance into a modulating role in perfusion quality. Apart from the synthesis of nitric oxide, S-nitrosylation (SNO) serves as the exclamation marks of the redox language. Moderate SNO modification of mitochondrial and cytoskeletal proteins maintains the fission–fusion balance and endothelial healing, while drastic nitrosylations of Drp1, NMDA receptors or actin-proteins incites cascades of excitotoxicity and apoptosis [48,49]. Therefore, the NO/SNO dynamics represents the narrow bandwidths of regulatory coherency and pathological incoherency. The secondary redox mediators serve to complete the grammar. The peroxiredoxins (Prx1-6) and thioredoxin TXNIP systems mediate extemporaneous detoxication of peroxides but, under conditions of hyperoxidation, turn to damage-associated molecular patterns (DAMPs), activating the NLRP3 inflammasome, establishing a bridge between oxidative imbalance and sterile inflammation [50]. Within the microcirculation, pericyte constriction and glycocalyx collapse act to establish nodes of critical failure. The oxidative surge, which occurs after reperfusion, leads to the elevation of intracellular Ca^2+^ and RhoA/ROCK signaling, subsequent to irreversible contraction of pericytes: fasudil shows a degree of reversibility [51]. Simultaneously, oxidative degradation of the endothelial glycocalyx liberates syndecan-1, hyaluronan, and heparan sulfate fragments into plasma, measurable biomarkers of microvascular injury. The loss of this layer disables shear sensing channels Piezo1 and TRPV4, disrupting neurovascular coupling. Perfusion imaging of capillary transit time heterogeneity (CTH) links these microvascular redox injuries directly to tissue outcome [52]. Neutrophil extracellular traps (NETs) further catalyze this ischemic chemistry. DNA–histone scaffolds carrying myeloperoxidase and elastase generate chlorinated oxidants (e.g., 3-chlorotyrosine) and locally reduce iron, precipitating ferroptosis in adjacent cells. PAD4 inhibition (GSK484) or DNase I degradation of NETs improves reperfusion and limits injury [53]. Thus, NETs are active redox reactors and not passive byproducts. The redox code is larger than the brain. Metabolites emerging from the gut microbiome, most notably succinate and trimethylamine-N-oxide (TMAO), short chain fatty acids, are structural dares to the systemic redox tone and immune cell polarization. A 2024 plasma metabolomic study identified oxylipins and acyl-carnitines as independent predictors of lesion size, confirming that stroke is a systemic redox disorder expressed with cerebral manifestations [54]. Temporal redox programs also present opportunities for reparation. Nrf2–Keap1 activation drives detoxification and angiogenic gene networks, HIF-1α promotes glycolysis and vascular remodeling, FoxO3a integrates cell cycle arrest with oxidative defense. These transcription switches are modified by peroxiredoxin and thioredoxin relays, empowering the idea that redox signals operate contextually. They are destructive early on but reparative when temporally repurposed [55,56,57]. To visualize the interconnected biochemical events that underpin this transition from oxidative injury to redox-guided recovery, Figure 2 summarizes the principal interaction pathways active during ischemic stress. Oxygen and glucose deprivation precipitates a rise in the NADH/NAD^+^ ratio, leading to mitochondrial electron flux reversal and the generation of reactive oxygen species (ROS). In sum, the interaction of influences produces what we now define as the redox code, the layered grammar of the electron (or redox) carriers, which delineate cellular destiny by empirically quantifiable thresholds. Each redox couple (NADH/NAD^+^, NADPH/NADP^+^, GSH/GSSG, BH_4_/BH_2_ and NO/SNO) serves as a clause in the biochemical vernacular to engage environmental stress signaling into the metabolic syntax. In this context, discretely tuned thresholds, such as the decrement in GSH/GSSG ratio from ≈10:1 or the oxidation of BH_4_ to BH_2_, define the transition from adaptive signaling to ferroptosis or nitrosative injury [58]. Hence, the “code” is mechanistic, not metaphorically full: a battery of coupled flowed electrons defining if oxidative episodes are reparatory in the face of cellular injury or develop into irremediable injury. It is by knowing these thresholds that enacts a redox interpretation of the ischemic paradigm to be construed as a programmable logic of survival and adaptation.

### 2.2. Network Reverberations and Translational Choreography

Redox disequilibrium retorts through neural networks. Parvalbumin interneurons, essential for gamma oscillations, are exquisitely redox-sensitive, such that their impairment promotes disorganized cortical rhythms and cognitive decline [59]. At the molecular level, nitrosylation and thiol modifications of NMDA and AMPA receptors impair the excitatory–inhibitory equilibrium, correlating directly with EEG slowing and the loss of functional connectivity [60]. The redox connectome concept, thus, interconnects molecular electron dysmanagement to macroscopic global network dysfunction. Importantly, these processes are now measurable. Quantitative biomarker end points include NADH autofluorescence, plasma GSH/GSSG, BH_4_/BH_2_, nitrosylated proteomes, cf-mtDNA, NET parameters and glycocalyx products, whereby advanced imaging modalities such as hyperpolarized ^13^C MRI, redox-PET and microdialysis during thrombectomy allowed for spatial mapping of these dynamics [61].

Therapies need to be choreographed as follows:Hyperacute (0–6 h)—RET- and ferroptosis inhibitors, DNase I, iron chelators [62].Acute (6–24 h)—fasudil, sulodexide, sepiapterin, arginase inhibitors [63].Subacute (1–7 days)—NRF2 activators, HIF modulators, mitochondrial biogenesis inducers [64].Chronic (>7 days)—rehabilitation under the auspice of redox biomarker measurement [65].

Future adaptive redox-guided trials should, thus, hierarchize patients by biomarker fingerprints (ferroptotic lipidomics, BH_4_/BH_2_ ratio, cf-mtDNA, NET score) and assign interventions commensurate with such molecular functional stages [66]. End points need to superadd those of NIHSS or mRS by inclusion of CTH, EEG connectivity and network recovery parameters, enabling true precision neuroprotection.

## 3. Mitochondrial Reprogramming and Redox Plasticity

Mitochondria determine the fate of the ischemic cell. Other than an ATP supply function, they receive structural, metabolic, interorganellar and intercellular signals promoting neuronal salvation or demise [67]. Here, we describe the main programmable control nodes—complex I/RET and succinate overflow, interoganellar consortia, dynamics, mitophagy, mPTP sensitivity, mitochondrial heterogeneity and messengers, cell–cell metabolic interchange, ferroptosis control—and tie these in to network function and timing of therapy (Figure 3).

### 3.1. Complex I Conformers and the RET Paradox

Ischemia tends to skew the NADH supply, whereas it inhibits the oxidative phosphorylation function, which serves to “insure” that complex I adopts an inactive, or “slipped”, conformation with unstable FMN pairing and increased electron leak on reperfusion [68]. Reperfusion of a hyper-reduced ubiquinone pool, thus, also favors reverse electron transport (RET), which results in superoxide pulses accounting for most of the ROS, which can commonly be detected early in succinate-enriched, ischemia-mimetic contexts [69].

RET is paradoxically both damaging and contextually adaptive, thus the concept of programmable redox control (e.g., selective RET inhibition in the absence of inhibition of forward electrons) [70,71]. Coupled with this, ischemia leads to succinate overflow: massive SDH-generated oxidation on reperfusion feeds the Q-pool to maintain RET, and succinate acting extracellularly via SUCNR1 also promotes inflammation and vascular leak; succinate spikes, linked via microdialysis at recanalization, are related to worse outcomes [72,73,74,75].

### 3.2. The Role of Organelle Consortia and the Dynamics–Mitophagy–mPTP Axis

Mitochondria act as part of the organelle consortia. ER–mitochondria interactions (MAMs) regulate Ca^2+^ transfer and lipid traffic, lowering the threshold for permeability transition; peroxisomal β-oxidation leads to peroxide release into a compromised detox environment, which synergizes with mitochondrial ROS to foster lipid peroxidation [76,77]. Nanotunnels connect mitochondria to shuttle proteins/metabolites and intact organelles, which may propagate ROS injury or mitigate it depending on the redox environment [78]. Morphologically, ischemia promotes fission via Drp1 at the expense of fusion via Mfn1/2–OPA1, fragmenting the network; incomplete PINK1–Parkin mitophagy allows the persistence of ROS-producing entities, furthering injury [79,80,81]. Ultimately, mPTP (mitochondrial permeability transition pore) opening—a Cyclophilin-D-sensitive, time-critical event—gives way to collapse of ΔΨm, arrestation of oxidative phosphorylation, and an obligate step towards cell death; therapeutically, time-dependent sensitivity to mPTP opening in the hyperacute phase should take place rather than treatment of a stable target [82,83].

### 3.3. Heterogeneity, Messengers and Systemic Interfaces

Single-organelle and spatial analyses show that mitochondria within a given neuron or glia have significantly different membrane potential, calcium buffering capacity, and redox capacity [84,85]. Some organelles are functioning normally, while adjacent organelles undergo permeability transition (opening the mPTP) and reactively produce ROS, resulting in local zones of oxidative stress that will spread to neighboring cellular compartments. In tandem, mitochondria release mitochondrial microRNAs (mitomiRs), peptides such as Humanin and MOTS-c, and oxidized cell-free mitochondrial DNA (cf-mtDNA), which lead to activation of the cGAS–STING and TLR9 receptor pathways [86,87]. Via these signals, subcellular heterogeneity in mitochondrial integrity becomes yet another modulator of inflammatory amplification and progressive cellular injury following ischemic insult.

### 3.4. Sex Differences, Immunometabolic Crosstalk and Intercellular Transfer

Sex steroids enhance mitochondrial antioxidant mechanisms, biogenesis (PGC-1α) and fusion (Mfn1/2, OPA1) to underpin the greater resilience seen in pre-menopause; loss of this tone post-menopause is associated with worse outcomes and hints at the need for androgen-aware redox strategies [88,89,90]. More broadly within the NVU, microglia come to favor glycolysis and inflammatory ROS, astrocytes program mitochondria to sustain neuronal function, and the endothelium employs antioxidative systems (e.g., PRDX4) to affix barrier function, collectively framing injury and repair as a multicellular metabolic choreography [91,92]. Strikingly, intercellular mitochondrial transfer occurs: astrocytes and endothelium donate organelles via LRP1-dependent paths to sustain neuronal energetics and junctional integrity, with congruent in vivo data and usage of tool compounds supporting drug-donatable hepatization of mito-donation [93,94].

### 3.5. DHODH–CoQ Ferroptotic Control, Network Coupling and Therapeutic Timing

Mitochondrial DHODH-CoQH2 is a defensive mechanism for protecting the cell from lipid peroxidation and ferroptosis by continuously reducing CoQH2 to neutralize lipid peroxyl radical formation [95,96]. When DHODH-CoQH2 is disrupted, especially when succinate-driven reverse electron transport occurs, mitochondrial redox homeostasis is disrupted, and the neuron is predisposed to ferroptotic death. Since mitochondrial function in terms of both electrical potential and redox flow impact how the neuron generates rhythmic activity and interacts with vascular elements, the disruption of mitochondrial function due to its inability to generate adequate rhythmic activity and interact with vascular elements disrupts the coherence of gamma rhythms, synchrony between theta-rhythms and gamma-rhythms, and connectivity at rest, referred to as the redox connectome [97]. Understanding this connection across multiple scales provides evidence for stage-specific therapeutic intervention: early modulation of RET and succinate metabolism; intermediate protection of mitochondrial function via regulation of the mPTP and mitophagy; and late bioenergetically based recovery using imaging and circulating biomarkers [98,99].

## 4. Cytosolic Redox Signaling and Pathway Crosstalk

The cytosol of the ischemic neuron is not merely a reservoir where oxidative overflow pours; it is a complex and dynamic stage where reactive molecules communicate biochemical information. In this context, oxidants do not serve as dispersed poisons but as dictated signals to determine whether the cell should attempt adaptation, engage in repair, or cross the line toward death. Throughout the last several years, a developing view has transformed oxidative stress from a type of nonspecific harm to a structured chemical language of cysteine chemistry, lipid rewriting, ion fluxes, immune infiltration, and even physical self-assembly [100]. In this section, we will describe how this complex code operates, and recognizing its translation may lead to completely novel translational grounds in ischemic stroke.

### 4.1. Reactive Oxygen and Nitrogen Species as Coded Messages

Molecules like hydrogen peroxide, superoxide and nitric oxide spin in the cytosol with specificity that contradicts their simplicity. They “tag” cysteine residues in kinases, phosphatases, and scaffolds and transform these thiols to sulfenic, nitrosylated, or glutathionylated forms [101]. Proteomic atlases have now mapped over 500 unique cysteine switches in ischemic neurons, each with its own reactivity threshold and microenvironment. ROS and RNS are not damaging, as they are also writing biochemical sentences: short-lived oxidation of phosphatases lets Akt stay phosphorylated and protective, whereas steady-state oxidation locks pathways into apoptosis [102].

NO sharply illuminates this duality. At low flux, endothelia-NOS-derived NO potentiates perfusion via guanylyl cyclase signaling. At high flux, the NO from iNOS and the superoxide react together to generate peroxynitrite, a reactive radical propagator that can ultimately lead to tyrosine nitration, DNA strand disruption, or permanent exhaustion of enzymatic defenses. So, the oxidant(s) are Janus faced in the cytosolic processes, as to whether they will be protective or harmful, depending upon amount or time [103].

### 4.2. The Collapse and Inversion of Antioxidant Defenses

Cytosolic antioxidants are a conglomeration—glutathione, thioredoxin, peroxiredoxins, glutaredoxins—and use will rely on recycling through NADPH. However, ischemia does not allow for recycling the subcellular enzymatic defense—especially true if the pentose phosphate pathway was blocked and NADPH was eventually depleted. The peroxiredoxins—maybe Prdx2 in neurons and Prdx6 in astrocytes—will be hyperoxidized and lag behind in the form of reactivity broke out bursts of stitches regulation [104]. Ironically, the peroxiredoxins released from the cell would enhance harmful inflammation via danger-associated molecular patterns proactively as microglial toll-like receptors. What was protection from inside is betrayal for someone outside [105].

This inhibition is not trivial. The oxidized peroxiredoxins alone are not biochemically reflective of stress but have enough uniqueness and view into redox with immunity that they could actually be found in plasma within hours of the start of a stroke [106].

### 4.3. Convergence with Regulated Cell Death Programs

Cytosolic redox events are not an isolated event; they are an opportune event amid an elaborate regulation of death. Here, in apoptosis, the oxidation of cardiolipin allows for the release of cytochrome c, and then caspases can be free of oxidative inhibition to cleave, and then all of the parts cleave. In necroptosis, ROS are used to stabilize RIPK1–RIPK3 to drive cells to rupture by MLKL. Finally, pyroptosis does involve oxidized thioredoxin to bind and, subsequently, TXNIP is released, to eventually bind inside to activate the NLRP3 inflammasome [107]. Ferroptosis relies on lipid peroxides (for example, 4-HNE), is not an island unto itself, and expands every adjacent cell death program connecting membranes to mitochondria and inflammasomes [108]. In 2025, co-mingling possibly different death pathways into a continuum of redox-governed fates when oxidative switches determine if the cell disassembles appropriately or catastrophically lyses or succumbs to a peroxidative lipid conflagration. From this unity may arise therapeutics that appear to re-wire multiple fates through a singular oxidative node. Lipid redox metabolism is a rewiring of the grammar of ferroptosis. If it is mitochondria donating electrons, it is cytosol re-writing lipid narratives [109]. Ferroptosis has made the leap from oppositional oddity to central ischemic target; at the axis of ferroptosis reactivity is GPX4, detoxication of lipid hydroperoxides, including FSP1 regenerating CoQ10 for radical scavenging. Both are ineffective under ischemia: GPX4 via depletion of glutathione, FSP1 via depletion to NADPH [110].

However, this is not a failure of ferroptosis as an (evolutionary) biogenesis program. Redox-sensitive enzymes, e.g., ACSL4 and LPCAT3, load membranes with arachidonic- and adrenic acids, such that peroxidation is anticipated. Plasmalogens, lipids that are abundant in neurons, serve as sacrificial antioxidants, a procedure to end the angst and (evolutionary) biogenesis programming that deprives plasmalogens [111]. Peroxidation leads to electrophilic products, including 4-HNE and isoprostanes, which act as second messengers to covalently modify Keap1 or NF-kB, thereby linking lipid oxidation to transcriptional fate. Thus, cytosolic ferroptosis is not a failure but rather a grammar of rewiring membranes; biosynthetic lipid substrate availability decides if you live or die. Redox–kinase–transcription relays and epigenetic echoes for oxidants supplement kinases to do the dirty work. MAPK/ERK cascade signals, transiently, protect against death but, if prolonged, activate apoptotic cascades. AMPK senses AMP/ATP ratios and cysteine oxidation and makes slight modifications to autophagic thresholds [112].

There are the transcriptional arbiters, which are even further downstream. Nrf2, anchored by Keap1 oxidation, activates the genes coding for antioxidants and induces astrocytes to export glutathione to give a little bit of neuroprotective redox fidelity. HIF-1alpha integrates hypoxia and oxidation and shunts cellular metabolism into glycolysis and angiogenesis but beware of edema from prolonged activation. NF-kB activated by redox unleashes an inflammatory gene program, generating debris clearance for surgical activation of systemic immunity [113]. Epigenetics is another layer. ROS alter DNA hydroxymethylation via TETs, modifying histone demethylation via the JmjC dioxygenases and m6A RNA methylation via either FTO or ALKBH5, all marking local redox status to the chromatin of targeted cells and transcripts. Meanwhile, the perfusion deficit induced glycolysis, and TCA dysfunction leads to lactylation and succinylation, ultimately the first physiologic hallmarks many steps further down the line, all changing the cell’s dialect [114].

Thus, the cytosol will do more than respond; it will provide a source of continuous epigenetic echoes of oxidative changes, such as changing the headspace of the sallmisbalanced-disrupted reserves [115].

### 4.4. Redox Condensates and the Architecture of Stress

A new paradigm emerged, in which not only did the redox signals alter chemistries, able to imprint topologies to the cytosolic surfaces. Either G3BP1 or TIA-1 oxidation nucleates the assembly of stress granules, while decimedia, oxidized G3BP1 can repurpose translation. Inflammasome prion-like mirror effects propagate across membranes stabilized by disulfide bridges and form cellular responses from peroxynitrite insult. Kinases can converge into redox-sensitive condensates to create catalytic droplets, enabling unyielding concentration and signaling amplification [116].

How this occurs is paradoxical but indicates that ischemic cytosolic redox involves many aspects of biophysical reorganization and biochemical reaction. Changing the dynamics of the condensate could represent an entirely new treatment intervention, targeting the structure rather than chemically mediated molecular [117].

### 4.5. Immune Infiltration as Redox Amplifier and Buffer

The second complication of the cytosolic oxidative phase is immune cells at the injury site, using myeloperoxidase to produce hypochlorous acid, are left with chlorinated proteins. Neutrophils (NADPH oxidases) produce surges of superoxide from NOX2 and release peroxides continuously from endothelial NOX4. Monocytes pose similar danger by producing extracellular traps (antibodies and MPO) that expose pre-existing redox-dynamic injuries [118].

Yet, single-cell transcriptomics show that not all immune cells will amplify tissue harm; on the contrary, some immune cells contribute to the amount of oxidative enzymes and likely compete with the oxidative state overall. In summary, stroke-induced inflammation is ephemeral (at least the treatment-induced component), and a mixture of amplifiers and defenders potentially leads to treatment outcomes based upon differentiation, not general inhibition [119,120].

### 4.6. The Ionic Symphony: Calcium, Zinc, and Copper

Oxidants and ions are involved in the feedback loop. ROS oxidation of your ryanodine receptors and IP3 causes your calcium release that constituted the reactants for the signal. In the presence of excess cytosolic calcium, calpain activation causes mis-programming of your oxidative enzymes, effectively obliterating defenses. Calcineurin (itself oxidized) can also affect NFAT signaling and others like metabolic activity to effect new gene expression [121].

Lastly, ischemia causes zinc release from metallothioneins. Metallo-zinc would halt mitochondria and generate too many freely reactive radicals. It will bind to and halt other oxidative proteins too. Copper adds yet another instrument to the mix, producing metallic oxidation states to lead to metalloproteinases activated and loss of vascular integrity with the neuroelements and tissue. Hence, with ROS, calcium, zinc, and copper, you have self-sustaining triads of ion dysregulation—redox disaster [122,123].

### 4.7. Translational Horizons: Coding Therapy in Time and Space

The realization that cytosolic redox is a coding system opens a door to the treatment code for the benefit of success. For example, you can no longer just use antioxidants in spurious time slot locations in the temporal lobe. But you may use timing or spatial timing in treatment and code for therapy [124]: Hyperacute timing (<6 h) reverses electron transport, succinate dehydrogenase modulation limits primary oxidative stress, and absence of NOX2, MPO or NETs will restrain elite suffering [13,125].Acute timing (6–24 h) increase FSP1, decreases ACSL4, or decreases NLRP3 inflammasome together, managing the lipid metabolism and curbing the low-magnitude inflammation [126].Subacute timing (2–7 days), stimulus-activated Nrf2, or HIF-PHD wisely suppressed, or integrated stress unified; boosts repairs and suppresses edema [127].Finally, chronic timing, redox-boosted E-exercise, dietary electrophiles, and rehabilitation will provide mechanistic functions of resilience [128].

Biomarkers provide the reflection of time x code. The biomarkers par excellence were oxidized peroxireduxins, or chlorinated tyrosines, and ferroptotic-armed oxylipins, circulating mitochondrial DNA, and forms made under formic acid oxidations. Methodologies such as HyPer7, ro-GFP2, and iNAP will provide instantaneous feedback on what redox achieved, while mass proof provides the better thiolicome [129].

In clinical translation, it should come as no surprise that adaptive trial systems are longitudinal in nature, where biomarkers will be code and reference anatomical and electrophysiological endpoints: capillary transit time heterogeneity, resting-state fMRI connectivity, as well as quantized EEG, which are par excellence the final act, the retelling of the story of redox regulation. There will be some cures, to be sure. Table 1 is meant to summarize the strong listing effects from the point of view of methodology, biological effects, and translation pitfalls for clinical and investigational centers.

Figure 4 supplies the synthesis by providing a converging framework of the therapeutic mechanisms. Integrating the molecular, physiological, and systemic in one translational continuum, from the inhibition of cellular ferroptosis to glymphatic-electrophysiological monitoring and redox-based metabolic regeneration, reveals how different experimental trajectories lead to a common therapeutic target for neuroprotection, being precise via the restoration of energetic coherence, vascular resilience, and neuro-immune stability during the acute, subacute, and reparative phases of stroke recovery.

## 5. The Neurovascular Unit and the No-Reflow Problem: Microvascular Redox, Barrier Failure, and Edema

While recanalization of an occluded cerebral artery is a foundational tenet of current stroke treatment, practical experiences demonstrate the apparent paradox of recanalization, resulting in no restoration of tissue perfusion. This deficit, long known as the no-reflow phenomenon, highlights the predominance of the capillary microvasculature and, by extension, all components that form the neurovascular unit (NVU). The NVU is not a passive framework but an incredibly dynamic, redox-sensitive collective of endothelial cells, pericytes, astrocytic end-feet, blood cellular elements, as well as extracellular components, which ultimately govern whether or not reperfusion leads to tangible tissue salvage [140]. In this section, we will aim to descriptively clarify how oxidative and nitrosative stresses modify every component of the NVU in the setting of ischemia reperfusion, how these factors facilitate capillary-level failure, and how they yield barrier dysfunction, edema, and long-term vascular network injury.

### 5.1. Capillaries as the Decisive Battlefield

At the capillary scale of assessment, microvascular flow is described much differently than in larger arteries, where microvascular flow is instead regulated by the driving forces, rheological aspects including the Fåhræus-Lindqvist effect, plasma skimming, and glycocalyx drag for tissue oxygen-consuming tissues. A microvascular redox imbalance impacts the mechanical driving force [141]. Two-photon microscopy and human OCT angiography have highlighted “capillary stalling”, a phenomenon of red blood cell cessation or stall, where after the proximal flow has been restored, capillary red blood cell stalling still continues. These stalls resulted in patchy hypoperfusion, which reflects an explanation of why perfusion remains ischemic, when arteries are widely open. Computational modeling indicates that capillary transit-time heterogeneity is a superior marker of oxygen extraction failure than mean perfusion with redox-mediated capillary dysfunction at the core of no reflow [142].

### 5.2. Pericyte Constriction as a Redox Chokehold

Pericytes, in particular, are metabolically separate neurons that are glycolytic and ischemia sensitive. Reperfusion comes with an oxidative burst, which depolarizes the power sources, leading to calcium release through oxidant-modified ryanodine and inositol triphosphate (IP_3_) receptors, leading to actomyosin contraction [143,144]. Oxidative genetics results in oxidative damage to the pericyte, which then activates polyd-ADP-ribose polymerase (PARP) for nad+ depletion, placing the pericytes in a ‘rigor-like’ state. Finally, nitration of the adhesion complex makes adhesion to the basement membrane hard and embeds the redox insult in the beef architecture of the vessel, which is similar to arteriolar smooth muscle [145]. This is likely irreversible. New studies have been proposed to treat pericyte-mediated constriction with Rho-associated protein kinase (ROCK) inhibition, endothelin blockade, and nanoparticle-targeted antioxidants that report their initial success in preventing/reversing pericyte-mediated constriction. Connectivity to optogenetic studies in nerves show that relaxing the mural cells improves capillary perfusion, thus supporting a method where mural cells may not be the only cause of occlusion, just the route to treatment [146].

### 5.3. Endothelial Dysfunction and Barrier Failure

The cerebral endothelium has a distinct redox sensitivity. Cellular outcomes may include the loss of endothelial nitric oxide synthase (eNOS) coupling when tetrahydrobiopterin (BH_4_) consumes the excess oxidative substrate, becoming a superoxide-generating enzyme, as well as vasomotor dysfunction and a pro-thrombotic state. Weibel–Palade bodies in endothelial cells release von Willebrand factor and P-selectin due to oxidative stress and increase the rolling of leukocytes and platelet aggregation [147]. Together, glycocalyx destruction exposes adhesion molecules and alters flow shear, which means the free-circulating remnant is seen in the plasma of patients in early NVU failure [148].

Junctional proteins experience oxidative modification (nitration of occludin and ZO-1; internalization of VE-cadherin), and MMP-9 degrades the basement membrane, allowing barrier leakiness. The junctions’ coupling, paracellular route to permeability, oxidative impairment of Mfsd2a, and nitration of caveolin-1 induce the release of large amounts of proteins and lipids transcytosis storms to the parenchyma and edema when junctions appear intact. Moreover, epigenomic studies add another layer: oxidative stress-mediated chromatin remodeling and regulatory RNAs, miR-155, lncRNA H19 induce an endothelium injury memory [149].

### 5.4. Astrocytic End-Feet and Polarity Collapse

Astrocytes regulate water homeostasis, neurovascular coupling via astrocyric end-feet. Under oxidative stress, aquaporin-4 channels relocalize in ways that derail unidirectional water flow, and cytotoxic edema occurs. Calcium is an oxidant, and milestones develop that release gliotransmitters in uncoordinated ways, uncoupling precision-based conductance to blood flow [150]. Astrocytes release via extracellular vesicles, information rich with oxidized lipids and microRNAs, and increasingly plasma biomarkers reflect astrocyte stress. Similarly, astrocytes use mechanisms of oxidative stress (glutathione efflux) and mitochondrial transfer to neurons and the endothelium to mitigate mitochondrial redox processes [151]. These mechanisms rely on oxidative stress overload. Glymphatic clearance is not spared. As astrocytic end-feet expand, the perivascular space collapses, resulting in toxic metabolite accumulation. Collectively, these mechanisms provide evidence of astrocytes’ redox disagreements in acute perfusion and chronic metabolic imbalances [152].

### 5.5. Thrombo-Inflammation as a Redox Amplifier

Circulating immune cell accumulation eats the microvasculature in a redox-sensitive inflammatory milieu. Platelets hyperpolarize, forming a mitochondrial phenomenon, causing enhanced accumulation and degranulation. Neutrophils exocytose their neutrophil extracellular traps (NETs), which contain oxidized histones and myeloperoxidase, causing entrapment of erythrocytes, increased lipid peroxidation, and flow disruption with entrapments [153]. Monocytes in the redox priming would initiate “trained immunity”, instructed to secrete tissue factor and cytokines to persist in microthrombosis. It explains the paradox of not restoring microvascular perfusion from an open artery from thrombectomy and/or thrombolytics. PAD4 blocker, MPO inhibitors and stabilizing NETs are new translational therapies for pathway targets, and cardiovascular clinical trials are emerging and will likely translate to stroke populations [154].

### 5.6. Erythrocytes as Redox Propagators

Erythrocytes are modeled as passive oxygen carriers; the mode of action is conceivably detrimental in NVU with oxidative stress. Oxidative cross-linking of cytoskeletal proteins such as spectrin and Band-3 impacts (i.e., impede) membrane passage through capillaries; exposure of intracellular membrane phosphatidylserine offers pro-thrombotic surfaces; and exocytosis of ATP blunted would cause a physiological vasodilator to blood vessels in shear stress [155]. Erythrocytes excise microvesicles posing as oxidized hemoglobin, and lipids previously characterized would injure the endothelium in a myriad of ways. Additional mechanistic presentations are the antioxidant defensiveness (i.e., peroxiredoxin-2, catalase, etc.), which are attenuated down signaling; new evidence may demonstrate that remnant mitochondrial particulate matter may indeed present not just as redox catalysts/amplifiers [156]. Novel emerging measurements of individual erythrocyte stiffness in microfluidic will be increased in stroke patients; with simultaneous and rapid stiffening, the proposed conceptualization remains, and they serve as functional connectors, both targeted for biomarkers in the future, as well as subsequent targets for therapeutic action [157].

### 5.7. Edema as a Redox-Governed Choreography

Edema is the ultimately outwardly visible end-point of microvascular obliteration. Ionic edema occurs in minutes, while Na+/K+-ATPase may not be able to work, also due to perozynitrite. The cytotoxic swelling of neurons and astrocytes compresses capillaries, magnifying hypoperfusion. Hours later, vasogenic edema develops, signaling junctional decoupling and caveolar transcytosis, and oxidative lipid peroxidation increases extracellular spaces [158]. Regional differences are unique; white matter has more sustained ionic dysregulation, and gray matter has rapid cytotoxic edema. Hormonal and age differences determine a patient’s outcomes. Estrogen prolongs aquaporin polarity among young females, while aged brains lose barrier integrity via diminished Nrf2 signaling. Sphingosine-1-phosphate receptor signaling is disrupted under oxidative stress, leading to loss of barrier integrity, demonstrating that edema is a redox-based choreography of overlapping cellular processes [159].

### 5.8. Translational Horizons for NVU Redox Biology

The transition of these mechanistic findings to clinical practice is emerging. Circulating biomarkers that reflect NVU-related stress are syndecan-1′s fragments, vWF/ADAMTS13 ratios, myeloperoxidase–DNA complexes, and oxidized erythrocyte vesicles. Techniques such as perfusion MRI, oxygen extraction mapping, and OCT angiography allow for the noninvasive visualization of capillary disease. Intra-thrombectomy microdialysis produces a real-time biochemical sample [160]. Treatments are increasingly targeting the NVU in phases, such as pericyte relaxants and BH_4_ donors in hyperacute, DNase and MPO inhibitors in early thrombo-inflammation, MMP-9 and S1PR1 modulation during acute edema, and Nrf2 activators or glymphatic enhancers during subacute recovery [161]. Looking ahead, redox-responsive nanoparticles that release antioxidants in ischemic microregions and CRISPR interventions targeting metabolic regulators such as Keap1, Mfsd2a, and AQP4 polarity are further examples of precision medicine in progress. Using machine learning methodologies that integrate biomarkers with imaging features can identify patients for complicated precision therapies, marking a shift towards personalized NVU therapeutics.

### 5.9. From Microvascular Collapse to Network Disintegration

The dysfunction of the NVU has consequences beyond microcirculation; it disrupts whole neural networks. Capillary stalls are a silence switch for fast-spiking interneurons, resulting in the breakdown of gamma oscillations on EEG. Functional MRI reveals loss of default-mode and frontoparietal networks in patients with marked microvascular dysfunction, and this correlates with an increased risk of post-stroke cognitive decline [162]. These findings place NVU integrity as a marker of global neurological outcome from hypoperfusion to beginning acute reperfusion therapy and chronic rehabilitation. Therefore, continuity of the capillaries and/or barrier integrity will not only save tissue but facilitate neural circuit priming for plasticity, leading to improved rehabilitation and secondary prevention [163].

### 5.10. Holistic Analytical Integration

Given all of these findings, oxidative stress is the logical unifying language of dysfunction at every aspect of the NVU, and from pericyte rigidity and endothelial uncoupling to astrocytic polarity failure, thrombo-inflammation, erythrocyte rigidity, and evolving edema, these each create the capillary grammar of no reflow. This grammar is why reperfusion therapies usually fail and provides a framework—mechanistic, biomarker-based, and therapeutic—for invariant arterial patency to be converted into meaningful recovery [164]. This proposal positions NVU redox reprogramming at the center of the stroke cascade; to understand the no-reflow problem will require recanalization of arteries and reprogramming the oxidative grammar of the microcirculation [165]. As presented in Table 2, the initial continuum for redo-reflux biomodulatory strategies can be donors such as ferroptosis inhibitors and enhancers of mitochondrial transfer, as well as state-of-the-art delivery of novel biomaterials. These retro-challenges represent in some ways how mechanistic rationales have already influenced translational strategies and that to adequately redox modulate reperfusion therapy, during hyperacute and subacute phases, ischemic strokes will be enhanced.

This table summarizes preclinical and early translational interventions (2023–2025) designed to modulate oxidative injury and recovery across multiple scales. Each entry specifies the mechanistic target, mode of action, supporting evidence, translational promise, and open limitations. Together with Table 1, this overview illustrates how redox biology is shifting from a descriptive paradigm to an actionable therapeutic framework.

## 6. Redox-Mediated Immunological and Glial Reprogramming in Acute Ischemic Stroke

The ischemic brain has long been more than the site of vascular incompetency; it has been recognized as the site of a constantly evolving immunological and glial battlefield. Reactive oxygen and nitrogen species are pivotal modulators of the myriad of resident and infiltrative cells, which extend beyond the acute phase of injury. Rather than caused solely by any factor that is solely toxic in nature, oxidative stress has now been recognized as a reprogramming signal for repair (when inflammation resolves) or when inflammation persists as chronic injury [179]. Here, we summarize the new advancements, particularly how redox imbalance takes its toll on microglia, neutrophils, monocytes, astrocytes and oligodendrocytes, and how microglial transformations propagate from alterations in their components to whole systems.

### 6.1. Microglial Plasticity Under Redox Pressure

Microglia are one of the first, if not the very first, cell types to sense acute (or chronic) ischemic stress and make dramatic changes to their transcriptome in the scope of tens of minutes after reperfusion. ROS stress causes metabolic reprogramming, where microglia shift from oxidative phosphorylation to glycolysis, which is consolidated by the activity of HIF-1α [180]. This metabolic shift comes before, resolving the microglial inflammatory cytokine secretion, and causes hyper-priming of microglia for sensitivity to activate the inflammasome. Mitochondrial ROS modulates the assembly of NLRP3; however, in conjunction with enhanced inhaled mitochondrial DNA segments, ROS functions as a DAMP to amplify inflammatory generated signals [181].

Spatial transcriptomics in 2024 suggests that oxidative microenvironments present in the peri-infarct cortex result in “microglia specific states”: 1. “ROS-dominance microglia” that are enriched for pro-oxidative NADPH oxidase and inflammasome genes, and 2. “antioxidant microglia” that were enriched in Nrf2 and lipid metabolism genes [182]. Though these microglia populations exist within millimeters of one another, their expression of distinct transcriptional programs illustrates microglial heterogeneity within spatially coded redox gradients [183].

Redox signaling leaves molecular scars long after. The presence of 8-oxoguanine at enhancers and redox-operated histone crotonylation (still not well studied, but some evidence points in this direction) appears to be associated with microglia “trained immunity,” which could put these cells at increased risk to have heightened responses to repeat ischemic or systemic insults [184]. Exciting new data suggest that neuronal cells may send signals to nearby microglia to dispose of distressed mitochondria, which is adding more oxidative load from the organelle and driving microglia to a more glycolytic, inflammatory program. Mitochondrial–microglia transfer of mitochondria may explain the lasting inflammatory tone after stroke [185].

Mitochondrial exchange has also been shown to be a two-way redox regulatory interaction between neurons and glial cells. When neurons experience damage from an injury, they release damaged mitochondria into the surrounding space; this is taken up by microglia, which then destroy them through transmitophagy (a form of autophagy). Conversely, when microglia or astrocytes become active in response to injury, they can provide healthy mitochondria to damaged neurons via tunneling nanotubes or extracellular vesicles to assist with the production of ATP [186]. The process of intercellular mitochondrial exchange, involving oxidative stress, calcium dyshomeostasis and mitochondrial depolarization, enhances ATP production in damaged neurons while reducing ROS levels. While poorly defined, the processes involved in intercellular mitochondrial exchange indicate that microglia act as both inflammatory sensors and regulators of cellular energy and can help to buffer the variability in the redox stability of the cell and aid in recovery after ischemia [187].

### 6.2. Infiltrating Leukocytes and Oxidative Traps

Neutrophils infiltrate very soon (within hours) and provide significant sources of oxidative stress. They produce neutrophil extracellular traps (NETs) as redox obstacles that are ornamented (furnished) with oxidized histones and myeloperoxidase (MPO), which are the capillary blockages and pro-thrombotic antagonists. In fact, data emerging from 2024 suggest NET fragments in the plasma enriched in oxidized proteins may predict hemorrhagic transformation after thrombolysis, indicating potential translational relevance to discovery [188].

Neutrophil heterogeneity complicates this picture. “Older neutrophils” are characterized by high levels of ROS, mechanistically more predisposed (greater likelihood) to produce NETs, and enhanced representation in the patient cohort with poor outcomes. Mechanistically, lipid peroxidation in neutrophils through ACSL4-dependent pathways lends to chromatin decondensation, and, thus, lends to NETs, which begs the question of there being a mechanistic association between ferroptosis-like lipid damage having NET formation [189,190].

Monocytes also undergo oxidative tenacity. Two types of inflammatory monocytes stabilize HIF-1α through elevated succinate and IL-1β levels, and reparative monocytes stabilize itaconate pathways to maintain ROS and promote angiogenesis [191]. Systemic issues such as diabetes and hyperlipidemia by augmenting oxidized lipids present in the circulation actually prime monocytes towards pro-inflammatory states and consequently elucidates how system-level vascular risk factors can also be assessed through the perspective inflammatory redox immune misalignments [192].

Last but definitely not least, T cells respond to oxidized signaling beyond that of innate immunity. New findings suggest that ischemic parenchyma CD8+ T cells experience mitochondrial dysfunction, lose cytotoxic activity, and now have some ROS-mediated cellular damage. However, regulatory T cells utilize redox thioredoxin pathways to protect themselves. This suggests that the redox environment of the lesion can dictate and control the balance between a cytotoxic and regulatory adaptive response [193].

In addition to the cellular components involved in this process, the connection between oxidative stress and immune response is facilitated through inflammatory cytokine signaling. Oxidative stress caused by mitochondrial ROS and oxidized lipids activates the NLRP3 inflammasome, resulting in the caspase-1-dependent cleavage and secretion of IL-1β and IL-18; both cytokines stimulate an increase in the number of recruited neutrophils and macrophages [194]. Activation of NF-kappa B stimulates the expression of NOX2, iNOS, and COX-2, thus increasing oxidative and nitrosative stress. Additionally, TNF-alpha and IL-6, secreted by macrophages and lymphocytes, influence the levels of activity of redox enzymes and antioxidant systems and promote either tissue destruction or repair [195]. Antioxidant systems controlled by Nrf2 and Sirtuin-1 have been shown to suppress NF-kappa B-driven cytokine transcription and, therefore, restore balance and possibly signal the transition from inflammation to resolution [196].

### 6.3. Astrocytes as Immune–Neurovascular Hubs

Astrocytes are not simply supportive cells; they are redox-responsive immune-shaping mediators. ROS can engage the STAT3 and NFAT signaling pathways, inducing the release of IL-6 and CCL2 that further promote leukocyte infiltration. At the same time, peroxynitrite can nitrate connexin-43 to destroy the astrocytic syncytium and impair ionic buffering, and there may be some ex-cytotoxicity. An exciting new dimension is astrocyte-derived vesicles [197]. Under oxidative stress, astrocytes release exosomes that contain oxidized phospholipids and pro-inflammatory miRNAs (e.g., miR29a) that, in turn, can eventually modify both endothelial and microglia cells. Under no or low-oxidative stress, astrocyte-derived vesicles contain miR-146a and neurotrophic factors to promote recovery [198]. In 2024, proteomic analysis revealed oxidized GFAP fragments in astrocyte-derived exosomes that could enter the circulation and correlate with stroke severity, implying astrocytes’ redox state is related to circulating stroke severity biomarkers. Astrocytes can complete antigen presentation [199]. The redox state allows for this in MHC-I and MHC-II peptide presentation, and this was demonstrated in 2024. This points to snorkel astrocytes as immune mediators of T-cell behavior during oxidative stress, whereby they transition from supportive glia to immune cells within the ischemic context [200].

Oligodendrocyte and myelin susceptibility oligodendrocytes have a high iron content and, together with the lipid-rich myelin, a high susceptibility to oxidative stress. Ferroptosis (GPX4 downregulation) and lipid peroxidation (through ACSL4) provide the cellular basis for oligodendrocyte death in stroke [17]. A single-cell lipidomics study in 2024 found oxidized phosphatidylethanolamines in oligodendrocytes bordering ischemic infarct cores. Oxidized myelin has the potential to be DAMPS and activate microglial inflammasomes that would worsen white matter injury [201]. Oligodendrocyte precursor cells (OPCs) are also subject to the same oxidative impairment, and even worse. ROS has been shown to downregulate Sox10 and Olig2, inhibit their maturation, and impact remyelination. Oxidized vesicles secreted by astrocytes carrying inhibitory miR secretion can also impede OPC maturation [202]. Astrocyte-derived “vesicular redox code” can likely determine when mesensomes/exosomes or E are identified as commitments for OPC maturation or if they remain in a quiescent state. Stress-induced loss of ceruloplasmin in oligodendrocytes could change iron homeostasis and increase availability for catalyzing Fenton chemistry, resulting in increasing lipid peroxidation. More importantly, white matter injury extends beyond a spike in infarct metabolism, often resulting in long-lasting cognitive impairment beyond what is presented with lacunar stroke [203].

Redox, Immune and Neuroplasticity Cross-talk

The ischemic space has created numerous forms of redox-based cross talk between glia/immune/neuro types. Microglia-induced ROS are unregulated wild/ROS-injured astrocytic aquaporin molecules, which make edema worse. From astrocytes to vasculature or directly delivered to microglia by vesicle exocytosis (oxidized sphingolipid), they activate microglial inflammasomes [204]. NET-generated oxidized proteins negatively modify OPC plasticity and incapacitate OPC maturation essentially, and/or lipids released from ferroptotic oligodendrocytes activate neuronal TLR4, subsequently restarting or stopping synaptic plasticity. This redundancy of mechanisms shows that redox stress is the hero of the narrative of cross talk among cells post-stroke. Crucially, these molecular releases are systemic dysfunction [205]. In vivo optogenetic and calcium imaging in 2024 showed that microglial oxidative pruning of dendritic spines is associated with the collapse of gamma oscillations. Connectomic fMRI analyses show that plasma markers of oxidative immune activity (NET fragments, oxidized vesicles) are compatible with the network dissolution of default-mode and frontoparietal networks, predicting cognitive decline after stroke. Thus, oxidative immune-glial re-programming is not only mechanistic but also predictive of neurologic outcomes in the long term [206].

### 6.4. Pathways to Clinical Translation

The translational space is quickly moving forward.
Microglia and astrocytes: Nrf2 activators such as dimethyl fumarate and synthetic triterpenoids shift phenotypes to antioxidant states [207].Neutrophils: PAD4 inhibitors, MPO blockers and DNase I are under trial in cardiovascular disease and, therefore, can be reformulated in stroke [208,209].Oligodendrocytes: Ferroptosis inhibitors (liproxstatin-1, ferrostatin-1) and iron chelators (next-generation deferoxamine derivatives) protect the integrity of myelin [210].Adaptive immunity: Modulating thioredoxin-dependent Treg survival gives new points of leverage [211].Biomarkers: Astrocytic and oligodendrocytic vesicles with oxidized proteins are being developed as plasma biomarkers for patient stratification [212].Nanotechnology: Redox-sensitive nanoparticles developed to release antioxidants only in ischemic microenvironments are now entering preclinical work [213].Gene editing: CRISPR directed toward Keap1-Nrf2 and Mfsd2a pathways will provide futuristic trials for tuning immune-glial redox signaling [214].

Artificial intelligence models that integrate these biomarkers with imaging features already demonstrate predictive ability for rehabilitation trajectories, signaling a future guided by redox signatures [215].

To sum up, if redox stress is the master conductor of immune and glial fates in acute ischemic stroke, it regulates microglial plasticity, biases neutrophil and monocyte programs, reorganizes astrocytic signaling, and specifies oligodendrocyte survival and remyelination. On a separate note, it redirects communication across glial, immune, and neuronal processes, making consequences observable in oscillatory coherence and network connectivity. By reframing oxidative stress not as collateral damage but, rather, a reprogramming signal, we present a therapeutic landscape, wherein the precise modulation of redox balance might shape immune-glial environment suffix with resilience and recovery.

## 7. Redox Control of Network-Level Dysfunction and Plasticity After Ischemic Stroke

Ischemic stroke is rarely limited to the infarct boundaries. What starts as a putative focal breakdown of energy metabolism and ion homeostasis quickly becomes a crisis of brain-wide communication. Networks that previously facilitated seamless hand-to-mouth coordination of movement, language, and cognition unravel, and, thus, the very substrate of recovery crumbles [216]. Oxidative stress is increasingly gaining recognition as the unexamined apparatus of this disorder, operating at different levels of understanding to connect small-scale molecular injuries with large-scale network breakdown [217]. Herein, we plan to outline how redox imbalance marks some synapses for deletion, disrupts oscillatory dynamics, elevates the putaminal connectome, and reduces white matter conductance, all while defining the biochemical cut-offs to allow or inhibit the brain’s capacity for adaptive plasticity.

### 7.1. Synaptic Integrity and Redox-Dependent Pruning

While traditionally viewed as the fundamental unit of plasticity, the synapse is a major place of susceptibility to redox imbalance. Oxidative modifications of scaffolding proteins such as PSD-95 and GluN2B, and presynaptic proteins like synaptophysin and SNAP25, destabilize the equipment of neurotransmission [218]. Recent imaging studies have gone even further, indicating that oxidized synapses are not simply dysfunctional but actively selected for removal. Microglia appear to sense these oxidized markers and mark these synapses for ingestion through complement pathways [219]. Oxidative damage also leads to the formation of the complement cascade that selectively eliminates these damaged or weakened synapses. Redox modification causes the binding of C1q, an oxidatively modified or misfolded synaptic protein such as PSD-95 or GluN2B, to synaptic membranes; subsequent cleavage of C3 leads to the formation of C3b/iC3b, fragments of C3 that act as molecular markers for the complement receptor 3 (CR3) found on microglia, leading to their recognition and engulfment of redox-modified synapses through a process similar to phagocytic ingestion by the immune system [220]. Continued complement activation due to either persistent oxidative stress from mitochondria ROS or chronic cytokine signaling (such as IL-1β or TNF-α) can convert normally adaptive synaptic pruning (refining connections) into destructive synaptic loss and network disruption after ischemic injury [221]. The end result is a nuanced but profound pruning of networks, which preferentially eliminate heavily connected nodes and, thus, remodels the framework of nodes that support global network operations. So, oxidative stress is a quiet author of both the surviving synapse and the dying synapse, establishing both local microcircuits and ultimately the sustained resilience of cognitive and motor circuits [222].

### 7.2. Oscillatory Dynamics and Network Coherence

If the synapse is the structural underpinning of connectivity, then oscillations are their dynamic enactment. These oscillations provide the rhythmic scaffolding, which allows distant regions to synchronize in time. Oxidative imbalance throws this rhythmicity into chaos. Nitration of GABAA receptor subunits diminishes inhibition, oxidation of pacemaker channels disrupts thalamocortical loops, and disconnection from astrocytes compromises potassium buffering. These disruptions collectively separate the consistent synchrony of oscillations [223].

Experimental work using optogenetics and in vivo calcium imaging was able to show that when microglia prune an oxidized synapse, rhythmic coherence was subsequently disrupted, in the prefrontal–parietal networks, specifically gamma rhythms. Clinically, this coherence manifests as loss of coherence on EEG related to working memory/executive function. The loss of gamma is not merely a corollary of injury; its absence presents as an early harbinger of chronic cognitive dysfunction, the emergence of the tangible cost of oxidative stress as the timing of our thoughts [224].

### 7.3. Connectome Disintegration Under Oxidative Stress

At the macroscopic level, a connectome can be indexed as the suffusion of axons, synapses and glia weaved into a network configuration. In the case of prolonged oxidative stress, that has gone awry. Astrocyte-derived microvesicles of oxidated lipids interfered with long-range signaling, and microglia-derived exosomes of NOX2 enzymes compromised cellular plasticity, even in contralateral sites [225]. Microvesicle-based signaling could account for the spread of injury beyond the location of the infarct. Thorough investigations of patients show that this can be verified. A large (multi-central) study conducted in 2025 found that circulating oxidative biomarkers (neutrophil extracellular traps, oxidized phospholipids, glial vesicles) predicted the rapid collapse of default-mode and frontoparietal networks, even correcting for infarct size [226]. Diffusion imaging has developed techniques to observe how white matter tracts, specifically the callosal fibers, can be acutely susceptible to lipid peroxidation and oligodendrocyte ferroptosis. With loss of the highest-degree vessels, a cascading disconnection can occur, whereby damage to only a few edges leads to the tenuously sustainable connectivity of entire networks [227].

### 7.4. Redox-Dependent Plasticity and Recovery Potential

Ultimately, oxidative stress is not simply a demolisher. Reactive species serve as signaling molecules at physiologic levels (to potentiate synapse, via ERK and CaMKII signaling) and can help with dendritic arbor refinement. But if this threshold is crossed, then the signaling will not promote plasticity occurrence, and the uncontrolled suppression of targeted pathways occurs. Thus, the trajectory of stroke recovery will be related not only to the presence of oxidative stress but the dosage and timing as well [228].

This has been verified clinically. Patients with a higher expression of antioxidant genes (i.e., Nrf2, SOD2, GPX1) show more reorganization across motor and language networks during things like rehabilitation. Patients with a high level of systemic oxidative load, even with similar structural preservation, show less functional recovery course. Extracellular vesicles can also shed light on biology: antioxidant vesicles correlate with dendritic sprouting; oxidized vesicles correlate with stunted outgrowth. This finding suggests that vesicular signals may, in the future, serve as valuable discriminators for who could be candidates for sequential intensive rehabilitation programs and who will need prior redox modulations [229].

### 7.5. Advanced Network Metrics and Computational Frameworks

Infarct volume metrics do not sequester oxidative ramifications, and it may be beneficial to pursue methods of analysis. Alongside methods of network neuroscience, we have ways to state this. Despite metrics of metastability, oxidant stress may obscure the large range of dynamic states and linger in the brain’s systematic interactions between integrated–segregated states of communication. In the case of patients’ EEG analysis, long-range temporal correlations were modified, and aperiodic slopes were amplified in specific patients with increased oxidative markers, implicating this slow ratchet as displacement from near-critical dynamics [230].

Very high orders of connectome harmonics, wide eigenmodes of sustaining default-mode and frontoparietal integration, were selectively decreased with redox constraints and reduced effectiveness of global ties. Computational models of oxidative microstructural injury directly impact the energetic constraints of configuration conditions of networks, decreasing controllability, and drive the system to non-optimal states. Network diffusion modeling can predict aging, where increasing the oxidative load leads to uniform dysfunction using structural backbones; now, removing the fruit of performance-based connectivity of networks, anywhere distant, partially intact [231].

In summary, these methods reconceptualize redox imbalance from molecular assault to a systemic change that will be the context for changes in brain dynamics and performance.

### 7.6. Translational Perspectives

Therapy is no longer targeting cells but networks. Antioxidant-loaded nanoparticles only deploying antioxidants in ischemic tissue established gamma synchrony that translated to behavioral recovery in preclinical models. Even transcranial alternating current stimulation, as a noninvasive application of neuromodulation, could be a target for application when in a lower oxidative state, as a way of interdependence with redox and circuit entrainment [232].

White matter-sparing treatments and the innovative use of ferroptosis inhibitors while combined with motor exposure can disrupt callosal and corticospinal scaffolding to allow the recovery networks. Closed-loop neuromodulation that changes stimulation involved with aperiodic slope and continuous EEG coherence becomes a potentially exciting new methodology of therapy within the brief state of plasticity.

Interestingly, an in-trial design process is changing the methods of process. No longer endpoints are infarct volume, as they are network related: gamma coherence; dynamic connectivity levels; controllability measures…now targeting derived. An adaptive platform process to use real-time stratified patient networks using a composer “Network Redox Index” uses vesicular genetic signals, NET fragments and EEG signals using EEG protocols, which may use an advanced model of biomarker-driven neuroplasticity [233].

### 7.7. Integrative Synthesis

Ultimately, redox stress is not benignly undesirable or ultimately hopeless. It is the chair and architect of networks’ fate to prune synapses, how to dethrone oscillations, when to collapse or orchestrate, or whether to embed in how long anyway-conceived plasticity of recovery would sustain. By cross-pollinating redox biology, network science, modeling, and biomarker processes together, we could develop a new vision. The point is not to draw in the antioxidant discovery at all but, rather, develop assumptions under redox stress, extinguishing, not emulating, learning and plasticity capabilities. From this definition, redox police can also be a network’s definition to restore coherence and sustainability across networks post-stroke.

## 8. Clinical Translation and Therapeutic Frontiers in Redox-Guided Stroke Care

While reperfusion therapies have been a revolutionary success in the management of acute stroke, outcomes in patients receiving reperfusion therapies are quite variable. Some patients may have a full recovery despite large infarcts, while others are left with considerable disability despite full recanalization. The variability in outcomes also reflects the limits of traditional recovery predictors of infarct volume and vessel patency, as these do not capture the more subtle biological mechanisms through which recovery potential and capacity occur [234]. Emerging evidence suggests that redox biology, through its regulation of network integrity and the potential for access to plasticity windows, may represent a better predictor of potential outcomes and target of treatment. Transitioning this information into the clinic will require not only a collection of therapies but a paradigm that encompasses molecular diagnostics, imaging developments, pharmacotherapeutics, neuromodulation, lifestyle management, and a complete overhaul of how clinical trials are conducted [235].

### 8.1. Biomarkers for Patient Stratification

Biomarkers are central to precision medicine, and in stroke, redox-based biomarkers are just beginning to demonstrate their worth as predictive biomarkers. Classic biomarkers for oxidative stress, like malondialdehyde, protein carbonyls, and 8-hydroxy-2′-deoxyguanosine, have been found to reflect systemic oxidative stress; however, they provide no spatial resolution or cell specificity [236]. More recent studies have begun to profile extracellular vesicles as vehicles of oxidized proteins and lipids from specific neural populations. Astrocytic vesicles with oxidized sphingomyelins appear to predict early white matter breakdown, while vesicles from oligodendrocytes with ferroptosis markers, like ACSL4, can provide evidence of the risk of demyelination [237,238]. Microglial vesicles with NOX2 enzymes display diffuse network disruption and center injury with the central degradation of lesions characterized in the articular response as areas of ischemic injury that do not correspond to direct foci of injury. While vesicles have been studied, neutrophil extracellular traps have also recently surfaced as a possible systemic biomarker. Their oxidized histones and DNA depict the junction of inflammatory oxidative stress, and multi-center trials yet again confirmed their usefulness in predicting cognitive trajectories [239,240,241,242]. They, along with proteomic, lipidomic, and metabolomic integration, generated unique “redox fingerprints” that distinguish patients by recovery phenotype with impressive specificity and sensitivity. Machine learning that employed these fingerprints performed better than clinical exam-scoring schemes, also supporting the potential personalization of biomarker-directed therapies will be on the horizon [243].

### 8.2. Imaging Windows into Redox–Network Interactions

While blood-based and CSF-based biomarkers provide meaningful molecular-level insights, imaging too has the novel potential to visualize how these travels occur at the systems level. Advances in diffusion MRI have already demonstrated the depth of the subtle structural fragility of the hub connecting tracts using pixel-based analysis, which showed early signs of degradation for patients with high oxidative load. Neurite orientation dispersion and density imaging provide another level of resolution, especially dendritic structure, and nearly perfectly characterize the oxidative pruning of brain tissue surrounding a stroke [244]. Research has moved from static fMRI analysis toward dynamic network interactions. High-oxidative-burden patients have a lower entropy of dynamic functional networks, meaning there are fewer transitions of network states for creating and integrating network states. In vivo electrophysiological readings may provide the best example for appropriateness and potential translatability [245]. Flattening the EEG aperiodic slope, loss of gamma coherence, and loss of phase-amplitude coupling have all been linked to oxidative stress and are, thereby, counter to bedside-accessible network vulnerabilities. Next, more sophisticated hybrid approaches are coming up to par, combining coupled AD-PET-MRI-based approaches using ROS-sensitive tracers for the size and shape of oxidative lesions, consistent with functional disruption, or joining up complementary modalities (likely in development) to enable redox–network coupling tagged with a suitable contrast agent [246]. Together, these imaging modalities are the cutting edge in the future of stroke, which is not just pinpointing functional breakdown but elucidating the redox–network relationship affecting young and old for brain repair [247].

### 8.3. Pharmacological Pathways to Redox Modulation

Pharmacology is also in re-emergence in where it moves from pan-purpose antioxidants all the way to harnessing specific and targeted redox modulation pathways that ultimately respond to the indifferent nature of reactive species as both ‘signal’ and ‘harm’. One prime example is activating the Nrf2 pathway using compound Dimethyl fumarate for multiple sclerosis. Screening products in vivo and quickly presenting with intractable stroke brain models has shown continued dendritic plasticity and axonal sprouting as an advancement [248]. In addition, new KEAP1 stabilizers and sulforaphane analogues performed similarly via selective Nrf2 stabilization. Ferroptosis inhibitors (liproxstatin-1, ferrostatin analogues, selenium-based) have a remarkable capability for maintaining oligodendrocyte health and preserving the integrity of myelinated fiber (all important for long-distance communication). Mitochondrially localized antioxidants (MitoQ and SkQ1) take it to the next level by specifically targeting sites of oxidative injury; this maintains oscillatory behavior while preserving the physiological ROS signaling necessary for health [249]. One of the most exciting strategies is intelligent nanoparticle systems, whose antioxidant cargoes only target ischemic tissue during hypoxia or acidosis, thereby selectively focusing on the brain and avoiding systemic uptake. Primate preclinical studies using catalase nanoparticles have demonstrated preserved corticospinal conduction and even improved motor recovery when used in conjunction with reperfusion therapies. Collectively, these pharmacological innovations represent a true paradigm shift; instead of eliminating oxidative stress (which is untenable), the goal is to calibrate oxidative stress within a redox range that fosters adaptive plasticity and prevents maladaptive overshoot [250].

### 8.4. Neuromodulation and Closed-Loop Strategies

While pharmacology works to adjust the redox tone from the inside out, neuromodulation attempts to reconnect network efficiency from the outside in. Noninvasive stimulation techniques are being extensively examined for motor and language recovery; transcranial alternating current stimulation (tACS) and transcranial random noise stimulation (tRNS) are both contingent upon the redox state at which stimulation occurs [251]. Prior animal studies using the same stimulation designs either facilitated motor assembly and electroencephalographic recovery or resulted in ineffective recovery, depending solely on the redox state that permitted synaptic potentiation. This is why closed-loop designs are so appealing, as stimulation during “redox-plasticity” timelines, indicated from behavioral performance EEG changes in coherence or slope, is favorable. In rodent animal models, timing-based neuromodulation interventions resulted in an almost two-fold increased rate of motor recovery, as compared to standard open-loop design interventions [252]. Indicated and presumably prior to this, invasive interventions such as deep brain stimulation within thalamic or basal ganglia nuclei, which are being evaluated for post-stroke cognitive dysfunction, will soon be suggested to be augmented or enhanced by incorporating real-time measurements of the redox state. Together, this information suggests a different therapeutic strategy, whereby neuromodulation is not uniformly applied but rather dynamically applied to the patients’ biochemical and network state to achieve the greatest potential benefit [253].

### 8.5. Rehabilitation and Lifestyle Interventions

Rehabilitation outcomes are increasingly appreciated as redox-dependent variables. The use of high-intensity training strategies, such as constraint-induced movement therapy and robotic-assisted therapy, yields the greatest results, tending to correlate to patients with low oxidative burden, while patients with high systemic redox stress may elicit less benefit than those with appropriate structural preservation. This suggests that biomarker-guided therapy intensity may maximize functional benefit. Further, concerns about biomarkers extend to lifestyle interventions [254]. Nutritional supplementation with glutathione precursors (e.g., N-acetylcysteine, glycine), mitochondrial cofactors, (e.g., nicotinamide riboside, alpha-lipoic acid) and polyphenolic antioxidants (e.g., curcumin analogues, resveratrol derivatives) may yield beneficial effects as adjunct therapies to standard care [255]. Exercise exerts a hormetic effect, with moderate- to high-intensity interval training producing transient increases in ROS that enhance endogenous antioxidant defenses, but excessive training can produce increased oxidative stress. Similar to therapy intensity, biomarker-guided exercise training may be feasible. Sleep is typically omitted, perhaps one of the most powerful vectors of redox balance. Sleep correlates with oxidative balance [256]. The impact of spindle-slow oscillation coupling, which is essential for memory consolidation, is upset by oxidative disbalance. The restoration of desynchronized slow-wave activity can re-calibrate disturbed redox homeostasis while also enhancing network plasticity. Together, these findings point toward the notion that recovery is not solely dictated by task practice but rather by the redox milieu of rehabilitation [257].

### 8.6. Clinical Trials and Precision Frameworks

The leap from concept to clinic ultimately rests on how trials are constructed. Traditional endpoints, such as infarct volume or NIH Stroke Scale, are poorly aligned with the biology of redox–network interactions. Future trials might rely on more sensitive and mechanistically relevant metrics, such as gamma coherence on EEG, entropy of dynamic functional connectivity, conduction parameters of hub tracts, or even vesicular redox cargo signatures [258]. Patient populations are heterogeneous, and the creation of a composite Network Redox Index—autoloading extracellular vesicle profiles, NET fragments, electrophysiological markers, and tractography indices—might be an amortized weapon to direct patients into those therapeutic arms [259]. Adaptive trial designs, as considered from oncology, can use umbrella, basket designs to also compare, in parallel, an Nrf2 activator, a ferroptosis inhibitor, nanoparticle delivery, and neuromodulatory therapeutics all from one adaptive platform. To do this, the harmonization of biomarker-strand processes, EEG preprocessing, and imaging pipelines across sites will be essential, as would a commitment to follow FAIR principles for data sharing. Ultimately, if this all pans out, stroke medicine will be the first neurology field with a fully biomarker-based, adaptive clinical trial framework [260].

The interactions, or lack thereof, of redox biology and network neuroscience present not just a batch of new tools but a new vision of stroke care. It treats oxidative stress not only as a downward trajectory of prognosis but a modifiable target that leads patients towards interventions overly determined by their molecular and network states. This aims not to “turn off” oxidative signaling, which is required for plasticity, but to “re-calibrate” it and achieve an adaptive range of signaling, which allows for the reorganization from a saved state but not a collapsed state [261]. The combined factors of biomarker-based stratification, multimodal imaging, precision pharmacology, redox-adaptive modulation, lifestyle optimization, and adaptive trial design might lead us to a time when stroke recovery is not only determined by the volumetric measurement of an infarct but the ability of networks to reorganize in the right biochemical background. While ambitious, and perhaps naive, this dream is becoming increasingly endorsed by data and arguably represents the most promising path toward individualized recovery [262].

## 9. Future Directions and Integrative Perspectives

Redox biology has the potential for significant advancements in stroke; it will be necessary to develop this molecular roadmap for reperfusion injury into practical, timely therapeutic options. In addition to being non-uniform, ischemic neuronal death is a series of sequential redox events that are triggered by RET at mitochondrial complex I and then amplified by the actions of xanthine oxidase, NADPH oxidase (NOX2/NOX4), and nitric oxide synthases [263]. The generation of reactive species, such as superoxide (O_2_•^−^), hydrogen peroxide (H_2_O_2_), peroxynitrite (ONOO^−^), and hydroxyl radicals (•OH), by these enzymes results in overwhelming levels of reactive species that exceed the capacity of antioxidants, leading to ferroptosis, necroptosis, and endothelial failure. Therefore, the next phase of research must establish a single “redox grammar” that describes the temporal, spatial, and systemic progression of oxidative stress and repair, by integrating all the mechanism-based nodes into a comprehensive model of redox signaling [264].

Therefore, the next stages of investigation will include moving away from simple descriptive models of stroke to identify specific, time-limited redox control points for the modulation of redox signaling—periods when mitochondrial succinate oxidation, lipid peroxidation, and nitric oxide signaling can be safely modified without compromising ongoing adaptation and recovery [235,265]. Redox-sensitive imaging techniques, including redox-sensitive PET, hyperpolarized MRI, and spatial metabolomics, may enable the targeting of specific times for intervention using RET inhibitors, ferroptosis blockers, or BH_4_-restoring agents during peak oxidative states. Such approaches will be enhanced through the integration of multiple types of omics studies at single-cell resolution, enabling an understanding of the differences in the oxidative–antioxidative status of neurons, astrocytes, and endothelial cells, and how Nrf2, sirtuin, and TET-dependent mechanisms alter gene expression following injury [266].

Attention must also focus on identifying the metabolic conditions (diabetes, obesity, aging) that modify the baseline redox tone of tissues through glycation, lipid oxidation and NAD^+^ depletion. Finally, the gut–brain–redox axis that is controlled by microbiota-derived short-chain fatty acids and kynurenine metabolism provides an additional layer of control over the oxidative load of the brain and the degree of inflammation priming. Therapeutic interventions designed to restore homeostasis of the microbiome and the metabolic system could potentially complement direct neurovascular interventions [267].

Computational modeling and bioengineering will provide the means to translate the findings into clinical applications. Digital twins constructed using artificial intelligence and incorporating molecular, electrical, and imaging data may provide predictions of the best time for treatment, while redox-responsive biomaterials and wearable biosensors could provide closed-loop therapy based on a patient’s current redox state. However, these technologies must be developed and implemented in a manner that ensures access and reproducibility worldwide, providing an ethical and equitable framework [268].

In summary, the future of stroke redox science will require connecting known enzymatic nodes (e.g., complex I, NOX, xanthine oxidase, GPX4, ALOX12/15, eNOS, and iNOS) to measurable redox biomarkers (e.g., BH_4_/BH_2_, GSH/GSSG, lipid peroxides, NET indices). Additionally, the field of redox research will need to transition from purely theoretical knowledge of the redox chemistry involved in stroke to a practical and clinically relevant application of redox biology that uses molecular timing to inform and guide clinical decision making [269]. Ultimately, redox research should evolve into a precision, feedback-guided, neurorestorative discipline that enables the conversion of reperfusion injury from a destructive event to a reversible biochemical state.

### 9.1. Temporal Choreography: Building a Redox Clock of Recovery

Healing after ischemia is not a linear or homogenous affair but is carried out in waves of oxidative and antioxidative spikes. In the first few seconds of reperfusion, prevailing peaks of mitochondrial superoxide emissions are in existence; during the parsec, similar peaks of leukocyte NADPH oxidase and peroxynitrite appear, whilst dated peaks of delayed nitric oxide and lipid peroxides express either a condition of damage or healing beginning. These peaks live in time-limited areas of biological evolution: gliosis, angiogenesis and circuit remodeling are examples, but these areas are also subject to elimination caused by circadian cycles of expression regulated by rhythms of melatonin and cortisol [270,271].

### 9.2. Spatial Atlases: Mapping Redox Heterogeneity Across Networks

Spatially oxidative stress appears plagued by heterogeneity: the metabolic hubs such as posterior cingulate, caudate and precuneus collapse during the initial phase of redox loads, whilst peripheral nodes fall to pieces in integration but maintain their structure [230,272]. Synchronous running of temporal redox clocks and spatial redox atlases, as dictated by redox-sensitive PET, hyperpolarized MRI and spatial metabolomics, may dictate the nature and time of investigative and therapeutical approach paradigms in such a way that antioxidants may be administered in defined oxidative peaks, so attempts at rehabilitation may be conducted during antioxidative recovery periods, and neuromodulatory manipulations may attempt the stabilization of fragile hubbed networks after [273]. Thus, healing is a choreographed redox ballet rather than a termination point of the journey.

### 9.3. Systematic and Epigenetic Convergence: Stroke as Systemic Redox Phenotype

Ischemic injury takes place against a landscape of systemic redox ecology. Diabetes, obesity, and age impose oxidative risk through glycation products, oxidative leak from lipids, and depletion of NAD+, while exercise, caloric modulation, and improvements in diet through polyphenols impose mitochondrial resistance [274,275,276]. Thereby, each individual presents with a unique redox phenotype that would suggest that therapy be stratified not by size of infarct but by systemic oxidative tone.

At the molecular level, the redox changes imprint themselves epigenetically. NAD+-dependent sirtuin activators, TET enzymes, and redox-sensitive histone deacetylases integrate the metabolic state with transcriptional reprogramming [277]. This oxidative asynchronicity results in epigenetic drift, in that genes important for recovery are diminished but do prolong the windows of plasticity through NAD+ replenishment or stimulation of sirtuin [278]. Ultimately, this places redox biology as the grand switchback of chronic adaptation, in that transient oxidative events are stored in the permanent genomic memory.

### 9.4. The Gut–Brain–Redox Axis

The gut microbiome has begun to be appreciated as a powerful architect of systemic redox homeostasis. The dysbiosis found in post-stroke states leads to lessened short-chain fatty acid producers, the emergence of hemolytic species, and ultimately lessening capacity for antioxidant function [279]. Butyrate and propionate act as histone deacetylase inhibitors, as well as increasers in transcription of antioxidant responses, while kynurenine pathway upregulation creates increased ROS and neuroinflammation. Microbiome transplantation methods have indicated that, through microbial reprogramming, one may increase systemic redox tone and recovery [280,281].

In addition to producing energy through metabolism, certain microbial populations can either positively or negatively influence an organism’s redox state. The production of glutathione by commensals (e.g., Lactobacillus and Bifidobacterium), and nicotinamide and folate metabolites, maintains NAD+/NADH homeostasis and supports mitochondrial activity [282]. Pathogens, on the other hand, produce lipopolysaccharides and secondary bile acids that actuate the TLR4/NF-kappa-B pathway, thus promoting the generation of ROS throughout the body. In addition to inducing oxidative stress and inflammation, short-chain fatty acid products such as butyrate and propionate also induce the transcription of genes encoding for antioxidants via the induction of Nrf2. However, when tryptophan is catabolized to kynurenine in dysbiosis, it leads to reduced levels of NAD+ and increased oxidative and inflammatory stress [283]. Therefore, the bidirectional communication between microbes and redox states defines a microbiota–redox–immune axis. Furthermore, the redox status of the intestine affects oxidative burden and neuroinflammation in the brain; conversely, cerebral ischemia changes gut permeability and, therefore, microbial populations. These are two components of a critical reciprocal feedback loop important for recovery from stroke and potential therapeutic modulation [284].

### 9.5. Single-Cell and Spatial Omics: Redox Landscapes at Cellular Resolution

Future probiotics, and even engineered strains, may, thus, function as bio-redox therapeutics with the benefits of precision medicine in the arena of microbial ecology. Concurrently, single-cell and spatial omics are presently resolving the redox landscapes at cellular detail. Neurons, astrocytes, oligodendrocytes, and microglia have different redox patterns, whereby neurons undergo bursts of mitochondrial ROS, astrocytes activate Nrf2, and oligodendrocytes undergo ferroptosis [285,286]. Mapping the “redox fingerprint” of these cell types allows for the targeted therapeutic delivery of ferroptosis inhibitors to white matter, mitochondrial stabilizers to neurons and Nrf2 activators to astrocytes. Stroke redox therapy, therefore, evolves from mass pharmacologic antioxidants to the targeted modulation of oxidative metabolism. Oxidative stress disrupts proteostatic homeostasis, linking stroke with chronic neurodegeneration. ROS interfere with endoplasmic reticulum folding, paralyze the ubiquitin–proteasome system, and freeze autophagy, leading to toxic protein accumulation [287]. This vicious cycle is mirrored by proteostasis collapse in neurodegenerative diseases such as Alzheimer’s and Parkinson’s disease [288].

Therapies restoring autophagic flux, upregulating ER chaperones, or preserving proteasomal function may produce cytoprotection in both acutely injured and chronically degenerating brains. The cloud of aging exacerbates this vulnerability. Senescent astrocytes and microglia have a pro-inflammatory, ROS-rich phenotype that inhibits regeneration [289]. Decreased NAD^+^, deterioration of mitochondria, and increased epigenetic drifting all reduce the window of plasticity. The merging of redox science with geroscience through senolytics or medical NAD^+^ restoration, therefore, provides a framework to create stroke therapy as a model of anti-aging intervention [290].

### 9.6. The Neurovascular–Glymphatic–Immune Continuum

At the interface of the brain and circulation, oxidative stress inflicts disproportionate levels of damage. ROS produced by the endothelial cells decrease the availability of nitric oxide and impair perfusion and the blood–brain barrier, while pericyte oxidative injury prevents reflow to microvascular [291]. Lipid peroxidation in astrocytic end feet disrupts the polarity of AQP4 and, thus, glymphatic clearance, resulting in edema and spatiotemporally congruous cognitive deficits [292]. Meanwhile, redox-modulated immune cascades activate NLRP3 inflammasomes and skew microglial activation toward M1 pro-inflammatory states [293]. Future treatments might integrate vascular, glymphatic and immune redox modulation, stabilizing the nitric oxide signaling pathway, preserving pericyte function, engulfing the NOX effects of inflammation, to bring the microcirculation back into line with the immune-resolving process.

### 9.7. Computational and Bioengineering Perspectives

Given the multidimensional complexities of redox dynamics, computational modelling becomes essential. Artificial intelligence might combine molecular, imaging and electrophysiological data into digital twins of stroke subjects, simulating recovery trajectories, testing potential interventions in silico and optimizing timing [294]. Graph-nested (or graph neural) networks may reveal latent redox–network connectome correlations, while temporal transformers juxtapose biological and oxidative oscillations, allowing for the prediction of optimal treatment windows. Transitions to clinical implications will require evolution in trial designs. Adaptive, biomarker-based structural designs could classify patients by redox network versus method of action and allow for multiple experimental approaches within an integrating structure [295,296]. Endpoints should look at coherence to coherence metrics (gamma rhythms), redox contents of extracellular vesicular components and immune cell oxidative signatures, linking outcome metrics yielded to the fundamental biology involved.

Concomitant to this, redox bioengineering illustrates tangible interfaces: redox-sensitive hydrogels releasing either antioxidants or stem cell products in response to the sense of oxidative cues; wearable biosensors allow for closed-loop neurostimulation dependent on the state of the local redox state; CRISPR-edited cells engineered to specifically neutralize local ROS [297]. Redox-predictive powers and regenerative engineering provide a template for adaptive redox-guided neurorepair.

### 9.8. Equity, Consortia, and Neuroethics

Finally, precision redox medicine should attempt to remain broadly available and ethically configured. Portable, low-cost assays of oxidative biomarkers, telephone-linked EEG systems for home use on a mobile phone basis, microfluidic urinary diagnostic kits for home use should democratize the possibility of redox monitoring at various levels of the health economy. International collaborations, such as those achieved in the Human Connectome Project, should synergistically create open redox–stroke databases, producing integrals of all possible omics, imaging and outcomes within the actuating research paradigm to permit collective fructification.

Yet, as we begin to have the power to modulate redox states in order to increase plasticity, therapeutic opportunities and implications will run foul of ethics: when does therapy become enhancement? Will the inequality of access to neuro-restoratives exacerbate health disparities? To answer these questions is no peripheral matter but rather a sine qua non for scientific, social legitimacy. The redox future will be a just futuristic one if the planning of the technology will be accompanied by enhancing custodianship. Nonetheless, the clinical progress of redox modulation is extremely nascent. Most drug interventions are afflicted with shortcomings in pharmacokinetic stability and therapeutic windows or the inability to differentiate adaptive from pathological redox signaling rapidly in time. Systemic antioxidants typically exhibit inadequate tissue specificity, and specifically targeted redox-active compounds (including ferroptosis inhibitors, NOX inhibitors and Nrf2 activators) require a precise timing and routes of administration, which generally will not lend themselves to experimental manipulation in the clinical setting. Furthermore, the extreme interindividual variability in metabolism, comorbidity and baseline redox states is detrimental to the propagation of the stratification of patients and accurate prediction of response to therapy. To further handicap patient stratification, the absence of standardized biomarkers between competing studies severely compromises the comparability and reproducibility of trial outcomes. To these disadvantages are superadded ethical and economic constraints against clinical advances, since sophisticated biosensors and means of redox imaging, as well as artificial intelligence-driven analytical science, require immense infrastructural developments. It may be that, whereas the clinical possibilities of redox therapies have made an intellectual advancement far beyond the classical field of antioxidants, their practical realization may be addressed accurately only by a careful regimen of temporal mapping of redox modulation, precisely individualized antidotal therapy and the incorporation of instantaneous feedback by biomarkers in adaptive clinical regimes.

In synthesis, the importance of redox biology lies at the intersection of molecular and systemic, temporal and spatial, biological and technological levels. The understanding and use of these dynamics—dependent on timing, mapping, individualization and ethics—will, hence, be the essence of the next generation of stroke recovery science.

## 10. Conclusions and Outlook

The key mechanisms found to be involved in ischemic brain injury are as follows:RET, occurring at complex I, is the major mechanism generating ROS during the initial phase of reperfusion;Ferroptosis, characterized by the inhibition of GPX4 via the lipid peroxidation of lipids dependent on iron, and subsequent loss of glutathione stores, is the major type of death for both neurons and oligodendrocytes;Failure of microcirculation due to constriction of pericytes, destruction of the glycocalyx, and decoupling of eNOS unite redox injury with vascular dysfunction;The amplification of inflammation through the activation of the NLRP3 inflammasome, subsequent release of pro-inflammatory cytokines via NF-kappaB, and formation of NETs link oxidative stress with the innate immune response;Both neuroimmune and metabolic adaptations, such as antioxidant defenses mediated by Nrf2 and sirtuins, transfer of mitochondria from glia to neurons, and redox modulation by the microbiota affecting recovery trajectories.

All these mechanisms describe a redox pathway of ischemic injury—a programmed sequence of the same oxidative stresses that damage cells can also be used to induce repair, plasticity, and regeneration when well timed and controlled.

We conclude this review, then, not with certainty but with curiosity. The tale of redox in stroke does not conclude with oxidative catastrophe but rather properly begins there and spreads outward through epochs of adaptation, timing, and survival. Ironically, what once appeared to be a story of arbitrary injury from radical assault now reveals itself to be a coded language of survival and repair, whereby the injured brain attempts to reconstruct its own platforms of recovery. Minutes and months apart, electrons themselves travel as messages—all to the end of informing upon and enacting whether tissues break down or restore. Redox, therefore, loses its status as a peripheral mechanism and becomes nothing less than the grammar of recovery itself: early oxidative burst can destroy, but those same forces may successfully rekindle, if shifted in time, angiogenesis, synaptic renewal, and metabolic equilibrium. The field itself remains in its adolescent phase. We lack a temporal chart of redox oscillations; a spatial atlas of oxidative heterogeneity and a cumulative integration of cosmic influences affects the metabolism and aging, such as the microbiome and stress. However, these lacunae are not failures but rather invitations. They demarcate the regions in which the next evolutions of discovery lie: in the syntheses and tensions of energy and time, structure and signals, to grow greater, to force destruction and repair.

The sciences of the future will no more measure redox, or absolute measure it, but they will interpret it as rhythm. The single cells and spatial omics will articulate it as signatures of oxidative oscillation with cellular precision; imaging and connectomics will resoundingly delineate their oscillation between figures; and the computational model may simulate the evolutionary regenerative architectonics, the digital twin of health. The trails of translation, which their newer ratios of time and scale will admit, must open a careful experimental check upon the regenerative processes that have no longer seventy-two breaths more than the allotted six, at present moment of ineluctable exhausting, but which must govern the stages of their ongoing subjugation to the damage heavy from reduction and oxidation. And yet, even after the essential course of data and ideas, different indeed than even on their profoundness, there is a unique truth: the very electrons that cause injury carry the primal reasons for nourishment. To know of their ebb and flow is to share the various points in the measure and rhythm of biological recreation between the dead and living, or idle, happy to be living.

We conclude, therefore, not with an ending but an opening. Within every stroke, there is a written tale and a certain writing. Redox balancing is not the endpoint of life’s order but its silent improvisation for renovation.

## Figures and Tables

**Figure 1 ijms-26-10835-f001:**
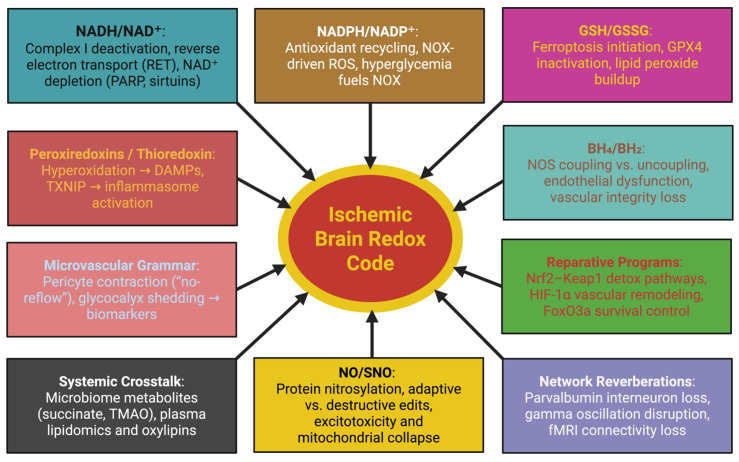
The redox code of the ischemic brain. A schematic representation of key redox couples (NADH/NAD^+^, NADPH/NADP^+^, GSH/GSSG, BH_4_/BH_2_, NO/SNO) and associated modules, illustrating their interactions and consequences across cellular, vascular, immune, systemic, and network levels.

**Figure 2 ijms-26-10835-f002:**
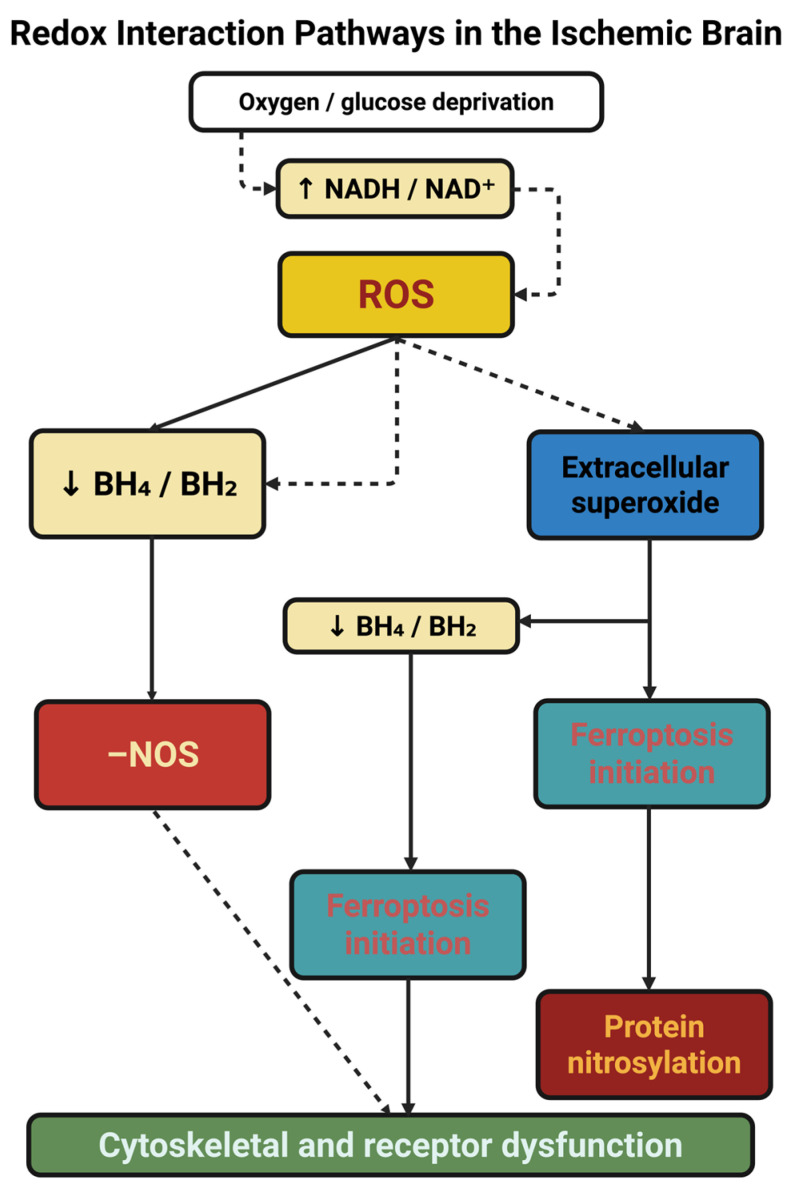
Oxygen/glucose deprivation acutely shifts the NADH/NAD^+^ ratio and promotes complex I transition to the deactive (D) state, enabling RET upon reperfusion and an early burst of ROS. In parallel, NADPH/NADP^+^ fuels NOX2/NOX4, augmenting extracellular oxidants; depletion of GSH/GSSG disables GPX4 and initiates ferroptosis; a reduced BH_4_/BH_2_ ratio uncouples eNOS, lowering NO bioavailability and increasing superoxide; and NO/SNO imbalance drives aberrant protein S-nitrosylation of receptors, cytoskeletal regulators, and mitochondrial proteins.

**Figure 3 ijms-26-10835-f003:**
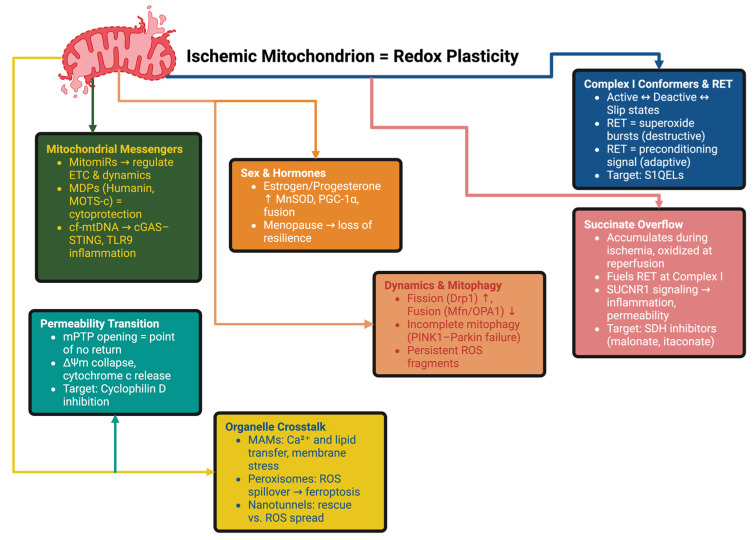
Mitochondrial reprogramming and redox plasticity. A schematic overview of ischemic mitochondrial responses across multiple layers. Key mechanisms include conformational flux of complex I and RET, succinate overflow, organelle crosstalk, altered dynamics and incomplete mitophagy, permeability transition, and release of mitochondrial messengers. Sex differences, immunometabolic reprogramming, and intercellular mitochondrial transfer further shape outcomes. The balance between destructive cascades and reparative pathways underscores mitochondria as programmable redox platforms and therapeutic targets.

**Figure 4 ijms-26-10835-f004:**
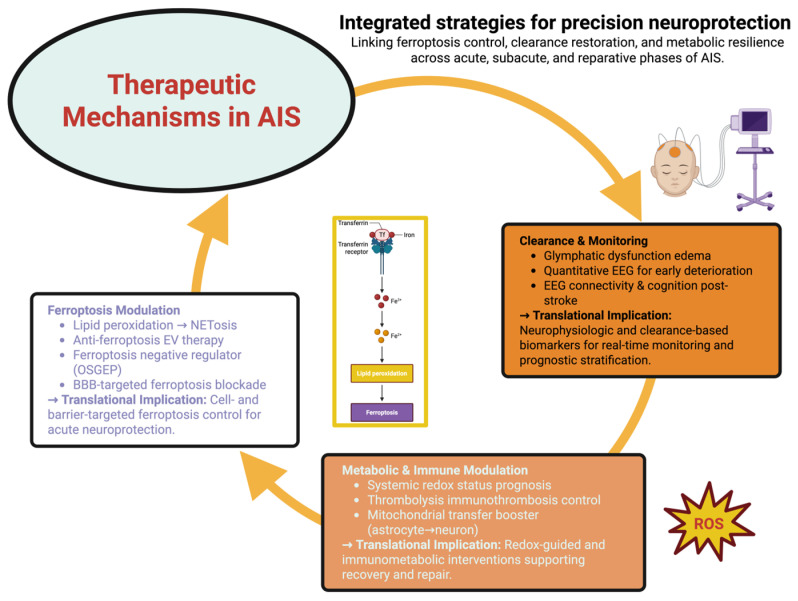
Therapeutic mechanisms in AIS. This schematic summarizes experimental and translational strategies targeting ferroptosis, glymphatic clearance, and metabolic repair in AIS. Ferroptosis modulation (left) prevents lipid peroxidation–driven NETosis and neuronal death through ferroptosis inhibitors, exosomal delivery, and BBB-targeted nanotherapies. Clearance and monitoring strategies (top right) use imaging and electrophysiological biomarkers—such as glymphatic DTI-ALPS and quantitative EEG—to detect edema and cognitive impairment early. Metabolic and immune modulation (bottom right) focuses on restoring systemic redox balance, regulating thromboinflammation, and promoting astrocyte–neuron mitochondrial transfer. Together, these approaches converge toward precision neuroprotection, linking acute ferroptosis control with subacute clearance restoration and metabolic resilience.

**Table 1 ijms-26-10835-t001:** Studies that advance the field of redox reprogramming in acute ischemic stroke. The table aims to synthesize recent experimental and clinical findings across molecular, cellular, and systemic domains, with particular focus on mitochondria-driven mechanisms, ferroptotic pathways, immune-redox crosstalk, microvascular integrity, and neurophysiological readouts. Each entry highlights the model or population studied, the specific redox markers or mechanistic readouts assessed, the core discoveries, and their potential translational applications, including therapeutic timing windows. Abbreviations: ALPS, Analysis along the Perivascular Space; BBB, blood–brain barrier; EV, extracellular vesicle; EVT, endovascular therapy; GPX4, glutathione peroxidase-4; GSH, glutathione; HMGB1, high-mobility group box-1; HRG, histidine-rich glycoprotein; LNP, lipid nanoparticle; LVO, large-vessel occlusion; MDA, malondialdehyde; NET(s), neutrophil extracellular trap(s); PSD, power spectral density; qEEG, quantitative EEG; tMCAO, transient middle cerebral artery occlusion.

Mechanism/Strategy	References	Models/Methods	Key Findings	Translational Implications
Lipid peroxidation → NETosis	[130]	Human neutrophils + platelets; in vivo validation	Iron accumulation and lipid peroxides trigger NETosis; small-molecule LP inhibitors suppress NETs	Platelets accelerate LP in neutrophils to induce pathogenic NETs; LP inhibition may serve as an anti-NET adjunct in AIS (acute)
Anti-ferroptosis EV therapy	[131]	MCAO mice	Engineered exosomes targeting M2 microglia prevent ferroptosis (↑GPX4, ↓lipid-ROS), reducing infarct volume	Cell-type-specific nanotherapies to preserve reparative microglia (acute–subacute)
Ferroptosis negative regulator (OSGEP)	[132]	tMCAO mice; OSGEP gain/loss; intranasal Fer-1/deferoxamine	OSGEP restrains ferroptosis; intranasal Fer-1 is immediately protective	OSGEP as targetable ferroptosis node; intranasal ferroptosis blockade in hyperacute AIS
BBB-targeted ferroptosis blockade	[133]	MCAO rodents; BBB-targeted lipid nanoparticles	Targeted LNPs deliver Fer-1 efficiently, enhancing neuroprotection vs. free drug	BBB-targeting makes ferroptosis inhibitors feasible for AIS therapy
Glymphatic dysfunction edema	[134]	MCAO mice + AIS patients (EVT); tracer assays; DTI-ALPS	Impaired glymphatic clearance contributes to brain edema; partial recovery post-recanalization	DTI-ALPS as biomarker of edema/clearance in AIS trials (hyperacute–subacute)
Quantitative EEG for early deterioration	[135]	LVO-AIS patient cohort; qEEG indices	qEEG detects early neurological deterioration beyond standard monitoring	Bedside electrophysiology to triage neuroprotectives/ICU resources (first 24–48 h)
EEG connectivity & cognition post-stroke	[136]	32 post-stroke participants (graded CI); gamma-band PSD; connectivity maps	EEG biomarkers classify cognitive impairment; gamma-band changes are discriminative	Noninvasive biomarker to pair with redox-directed rehab (subacute)
Systemic redox status prognosis	[137]	AIS patient cohort; serum GSH & MDA	GSH/MDA ratio predicts early post-stroke outcomes	Peripheral redox markers to enrich and stratify neuroprotection trials
Thrombolysis immunothrombosis control	[138]	AIS patients (proteomics, validation cohort); photothrombosis mice	tPA raises HRG; HRG dampens NETosis and reduces hemorrhagic transformation risk	HRG as biomarker/adjunct to extend safer thrombolysis windows
Small-molecule booster of mitochondrial transfer	[139]	OGD/R models; MCAO mice; chrysophanol treatment	Chrysophanol accelerates astrocyte→neuron mitochondrial transfer, improving neuronal survival	Tool compound suggesting druggable mito-donation pathways (hours–days)

**Table 2 ijms-26-10835-t002:** Therapeutic strategies targeting redox reprogramming in acute ischemic stroke.

Therapeutic Strategy	Mechanistic Target	Mode of Action	Preclinical/Clinical Evidence	Translational Potential	Limitations/Open Questions	References
Ferroptosis inhibitors (Fer-1, Liproxstatin-1, Deferoxamine, OSGEP upregulation)	Iron-driven lipid peroxidation, GPX4/GSH depletion	Block lipid ROS accumulation; restore GPX4 axis	Intranasal Fer-1 protects MCAO mice; OSGEP overexpression suppresses ferroptosis; BBB-targeted LNP-Fer-1 improves outcomes	First-in-class neuroprotective adjunct to reperfusion	BBB delivery challenges, optimal timing, systemic iron interactions	[166]
Mitochondrial transfer enhancers (Chrysophanol, LRP1–ARF1 lactylation axis modulators)	Neuronal bioenergetic collapse	Promote astrocyte-to-neuron mitochondrial donation	Chrysophanol accelerates mitochondrial transfer in MCAO; LRP1 knockdown ↑ transfer, ↓ injury	Metabolic rescue in penumbra; druggable axis for precision therapy	Integration with existing therapies, long-term safety, dosing optimization	[139,167]
Nanoparticle-based delivery (BBB-targeted LNPs, ROS-responsive nanocarriers)	ROS surge, antioxidant instability	Encapsulate antioxidants (Fer-1, edaravone, melatonin) for controlled lesion-site release	BBB-targeted Fer-1 LNPs outperform free Fer-1 in rodents; ROS-cleavable carriers validated in I/R	Overcome pharmacokinetic failure of antioxidants	Manufacturing complexity, immune clearance, scalability	[168,169]
Exosome therapies (engineered EVs targeting M2 microglia)	Microglial ferroptosis, immune-redox imbalance	Deliver miRNAs/GPX4 stabilizers to preserve M2 phenotype	Engineered exosomes prevent M2 ferroptosis, ↓ lipid ROS, ↓ infarct size	Cell-specific nanotherapy with natural biocompatibility	Standardization, off-target effects, GMP manufacturing hurdles	[131,170]
Immune-redox modulators (IL-1β blockade, CCR2/5 inhibitors, HRG augmentation)	Maladaptive immune memory, NETosis, BBB injury	Block IL-1β to prevent trained immunity; HRG suppresses tPA-induced NETosis	IL-1 blockade prevents cardiac fibrosis in post-stroke mice; HRG reduces hemorrhagic transformation in AIS patients treated with tPA	Extend thrombolysis safety, systemic protection	Risk of immunosuppression, patient stratification, timing window	[171,172,173]
Antioxidant cocktails with metabolic coupling (Lactate + NAD^+^ boosters, CoQ10 analogs)	Redox–metabolic axis disruption	Buffer ROS while supporting ATP/lactate metabolism	NAD^+^ precursors synergize with lactate shuttle in MCAO models	Personalized metabolic–redox therapy	Human dosing strategies, feasibility of combination regimens	[174]
Neurotechnology-based redox control (Vagus nerve stimulation, tDCS)	Excitotoxic–oxidative interplay	Modulate neuronal excitability to rebalance ROS	VNS reduces oxidative markers, improves plasticity in rodents; pilot tDCS data emerging	Noninvasive, scalable adjunct	Need for biomarker guidance, mechanistic precision	[175,176]
Systemic interventions (Microbiome modulation, exercise-induced ROS hormesis, caloric restriction mimetics)	Systemic oxidative background, gut–brain–redox axis	Modify microbiota; induce adaptive ROS hormesis	Exercise and microbiome-targeting shift redox tone in stroke-prone rodents; early human pilot data (2024)	Accessible, low-cost adjunct for prevention and recovery	Interindividual variability, standardization across populations	[177,178]

## Data Availability

No new data were created or analyzed in this study. Data sharing is not applicable to this article.

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
