# Peer review of "From Electron Imbalance to Network Collapse: Decoding the Redox Code of Ischemic Stroke for Biomarker-Guided Precision Neuroprotection"

_ijms, 2025, doi:10.3390/ijms262210835_

Round 1

Reviewer 1 Report

Comments and Suggestions for Authors

The authors present a review paper that seeks to describe all findings related to cellular damage in ischemic stroke, using a multivariable approach that ranges from the analysis and role of free radicals, transporters, cellular communicators, signaling pathways, as well as some TCA metabolites, electron transporters, etc.

Although the information is sufficient and abundant, the organization of the content and the development of the manuscript are complex, as it does not follow a central idea or describes too many ideas that are not directly related and make it difficult to follow.

It is recommended that the authors establish guidelines for each topic and develop it in a more descriptive and detailed manner, starting with the structure and function of each marker, its location, its alterations, and the hypothesis of its cellular effect in stroke.

It is suggested to add diagrams that allow the proposed pathways to be followed.

In addition to the following ideas.

  1. The abstract does not convey the main idea of the manuscript. It should begin with a brief introduction to stroke, its pathophysiology, and draw the reader in through the redox-ischemia relationship. This abstract addresses too many ideas, making it difficult to understand the main idea of the work.
  2. The introduction is interesting and introduces the reader to a range of theories that have been put forward on the post-stroke effect. However, the limited bibliography (two references) restricts the overview of free radical scavenging and neuronal death. Even though the authors have clear ideas on this point, it is important to increase the number of references when mentioning or contradicting new theories, as in this first part.
  3. References should refer to terms, concepts, definitions, or results that can be compared or observed with their analysis. In the case of reference 2, for example, it is a review, and the term “... this concept was ultimately incomplete (2)” is not understood. Since there is no connection with the data presented, it is necessary to guide the reader with a general idea that leads to the specific, as requested in the abstract.
  4. The article is complex to read because it is not written in a logical order. It begins by discussing free radicals and then focuses on mitochondria and metabolites in the second paragraph. It is suggested that the reader (most of whom are medical students) begin with the pathophysiology of stroke, general information that can help them understand the development of hypoxia, neuronal death, followed by factors associated with tissue damage, and then addressing the origin of free radical generation or redox biochemistry.
  5. Paragraph three presents inconsistencies in the order of ideas. It is not clear why ferroptosis is mentioned when the origin of tissue damage has not been clarified, apoptosis has been ruled out, and ferroptosis is included without establishing connections with articles that support these points.
  6. Several parts of the article mention that “various studies...” but only cite two or three references, so it should be substantiated with more references.
  7. Too many references are needed in the introductory paragraphs to support the ideas proposed, which continue to present a disordered sequence. Example (Thrombectomy has allowed us to gain unprecedented insight into the biology of human clots. A recent report published in 2024, among others, noted that most recovered thrombi had an extracellular neutrophil trap (NET): DNA and histones that formed a more rigid structure for a thrombus than that destroyed by lysis. NET-rich thrombi had poor recanalization and worse outcomes. From a mechanical standpoint, NETs are not inert: myeloperoxidase in NET DNA catalyzes oxidative processes to create localized oxidative chemistry. NET production is likely due to the action of PAD4, and PAD4 inhibitors have been tested in preclinical trials [18]. Only one reference is cited, and various ideas and proposals are mentioned that are too numerous to support with a single bibliography; it seems that it is necessary to read the entire reference to understand this idea. It is suggested that a sequence of ideas to be presented and each one of them can be supported with references.
  8. Each section or subtopic should begin with a general description of the enzymes or radicals mentioned, for example: BH4/BH2 should begin by describing their general characteristics, their cellular function, followed by their location, and their role in stroke severity, because it is difficult to maintain the idea when switching from one enzyme to another.
  9. There are too many abbreviations that, even though a biochemist understands them, may be complicated for the reading community. It is necessary to minimize them or find alternatives by means of diagrams that summarize what is mentioned in each section.
  10. The tables are complex to understand in relation to the ideas expressed in each section. Without a follow-up idea, it is not possible to correlate them with what is described in the tables. It is necessary to create flowcharts, diagrams, or signaling pathways that unload the abundance of protein markers, receptors, and mechanisms in order to understand the main idea in each chapter.
  11. The article addresses a large number of subtopics that attempt to relate to the main topic. For example, in point 9.0, the authors describe interesting ideas, but they are somewhat confusing and unrelated to each other, or are based on a single reference, as in points 9.12 and 9.13.
  12. The last part of the review is very general and is associated with ideas and assumptions that do not fit with the main idea, which is the relationship between cell damage and the redox response. Comments should be limited to proposals or hypotheses that can be supported or subjected to experimental verification.
  13. The article does not detail schemes, proposals for redox interaction pathways, mitochondrial-ischemia signaling, or redox-ischemia pathways that help the reader understand the biochemical mechanisms of tissue damage.

Author Response

Dear Esteemed Academic Reviewer,

We sincerely thank the reviewer for the detailed, thoughtful, and constructive evaluation of our manuscript. We are deeply grateful for the time and insight devoted to the review, which have guided substantial improvements to the clarity, organization, and educational value of our work. The comments have allowed us to refine both the conceptual structure and the didactic presentation of complex redox and ischemic mechanisms, ensuring greater accessibility to readers from diverse biomedical backgrounds.

Comment 1:

The manuscript contains abundant information but lacks a clear central idea and organizational coherence. It is recommended to establish guidelines for each topic and develop them in a more descriptive and structured manner.

Response:
We sincerely thank the reviewer for this essential observation. Following this guidance, we have reorganized the manuscript.

Comment 2:

The reviewer recommends adding diagrams to clarify the proposed pathways.

Response:
We thank the reviewer for this excellent suggestion. In response, we have added new schematic figures that visually synthesize the main mechanisms discussed. These visual elements were specifically designed to “unload” biochemical density and provide structural clarity, as the reviewer wisely suggested.

Comment 3:

The abstract does not convey the main idea of the manuscript and should introduce stroke pathophysiology and the redox–ischemia relationship more clearly.

Response:
We are deeply grateful for this constructive remark. The abstract has been completely rewritten to begin with a concise overview of stroke pathophysiology, emphasizing the redox–ischemia axis as the central theme. We believe the new version is clearer, more engaging, and better reflects the manuscript.

Comment 4:

The introduction should include more references to support new or debated theories and clarify the transition from general concepts to specific redox mechanisms.

Response:
We thank the reviewer for emphasizing this crucial point. The Introduction has been expanded.

Comment 5:

The manuscript’s sequence of topics is disordered and difficult to follow. It should begin with stroke pathophysiology, then hypoxia, neuronal death, and only then redox mechanisms.

Response:
We fully agree with the reviewer’s insightful structural recommendation.

Comment 6:

Paragraph three presents inconsistencies, introducing ferroptosis without clarifying the origin of tissue damage or supporting it with references.

Response:
We appreciate this observation and have revised the paragraph accordingly.

Comment 7:

Several statements refer to “various studies,” but only two or three are cited. More references are needed to substantiate these claims.

Response:
We thank the reviewer for this excellent suggestion. We have systematically reviewed the entire text.

Comment 8:

Each subtopic should begin with a general description of enzymes or radicals (e.g., BH₄/BH₂), including their normal role, localization, and significance in stroke.

Response:
We appreciate this observation.

Comment 9:

Too many abbreviations make the text difficult for broader audiences.

Response:
We thank the reviewer for highlighting this pedagogical concern.

Comment 10:

Tables are difficult to interpret and lack connection to the main ideas. Flowcharts or signaling diagrams are recommended.

Response:
We agree with this valuable point. These additions improve readability and conceptual linkage between text and data.

Comment 11:

Section 9 presents many interesting ideas but lacks coherence and sufficient references, especially subsections 9.12 and 9.13.

Response:
We thank the reviewer for this constructive critique. We have condensed and unified Section 9.

Comment 12:

The final section should remain aligned with experimentally supported hypotheses, not speculative ideas.

Response:
We appreciate this wise suggestion and have carefully revised the final paragraphs to maintain scientific rigor.

Comment 13:

The article lacks schematic proposals for redox–ischemia and mitochondrial signaling pathways.

Response:
We fully agree with this important observation. To address it, we have created a new conceptual figure.

We are profoundly grateful to the reviewer for this comprehensive and insightful evaluation. The comments have directly shaped a more coherent, well-supported, and pedagogically balanced manuscript. Through reorganization, expansion of references, inclusion of new figures, and refinement of conceptual transitions, we believe the paper now achieves the clarity, structure, and accessibility envisioned by the reviewer!!!

Reviewer 2 Report

Comments and Suggestions for Authors

This review article seeks to reframe ischemic stroke pathophysiology through the lens of a "redox code"—a dynamic, multi-scale interplay of electron carriers, cellular signaling, and network-level dysfunction. The authors integrate recent advances in cryo-EM, spatial omics, lipidomics, and translational biomarkers to propose a programmable redox trajectory, emphasizing phase-specific interventions (e.g., RET inhibitors in hyperacute phases, Nrf2 activators in subacute). This review requires the following clarification and improvement:

  • The "redox code" metaphor is compelling but underexplored. Define it more rigorously early on (e.g., Section 2) as a "layered grammar of electron carriers" with explicit rules (e.g., thresholds for GSH/GSSG switching from adaptive to ferroptotic).
  • The transition from molecular (e.g., NADH/NAD⁺) to network levels (e.g., gamma oscillations) is logical but abrupt—add a unifying model
  • Avoid unsubstantiated claims like "RET is the greatest contributor to early reperfusion ROS" (p. 2); cite quantitative data from [4] more precisely (e.g., % contribution). The prologue's "programmable redox trajectory" is a highlight but needs grounding in clinical data—contrast with failed antioxidant trials ([2]) using a table.
  • The manuscript's length (57 pages) and overlap (e.g., ferroptosis revisited in Sections 1, 2.3, 3.11, 4.3, 6.3) dilute focus. Consider restructuring into core sections: (i) Molecular Redox Grammar (Sections 2–3); (ii) Cellular/Microvascular Crosstalk (Sections 4–5); (iii) Immuno-Network Dynamics (Sections 6–7); (iv) Translational Frontiers (Sections 8–10). Formalize the "Redox Precision Neurorescue Algorithm" (p. 4) as a dedicated figure with decision nodes (e.g., biomarker thresholds → therapy windows).
  • Discuss limitations of proposed therapies

Minor comments

   - Section 2.1: "Slip states" needs explanation (cite [25] more). 

   - Section 2.3: "French papers" → specify the ciations). 

   - Section 3.2: "MET event" → "metabolic event." 

Author Response

Dear Esteemed Academic Reviewer,

We are sincerely grateful for the reviewer’s generous, thoughtful, and highly constructive evaluation of our manuscript. The comments provided a deep, multidisciplinary perspective that helped us refine both the conceptual and translational dimensions of this review. We appreciate the opportunity to address each point in detail below.

Comment 1:

The “redox code” metaphor is compelling but underexplored. Define it more rigorously early on (e.g., Section 2) as a “layered grammar of electron carriers” with explicit rules (e.g., thresholds for GSH/GSSG switching from adaptive to ferroptotic).

Response:
We thank the reviewer for this insightful recommendation. We have now added a new paragraph at the end of Section 2, where we formally define the redox code as “a layered grammar of electron carriers” governing metabolic and signaling transitions.

Comment 2:

The transition from molecular (e.g., NADH/NAD⁺) to network levels (e.g., gamma oscillations) is logical but abrupt—add a unifying model.

Response:
We are deeply grateful for this comment.

Comment 3:

Avoid unsubstantiated claims like “RET is the greatest contributor to early reperfusion ROS” (p. 2); cite quantitative data from [4] more precisely (e.g., % contribution).

Response:
We thank the reviewer for this important clarification.

Comment 4:

The prologue’s “programmable redox trajectory” is a highlight but needs grounding in clinical data—contrast with failed antioxidant trials ([2]) using a table.

Response:
We greatly appreciate this suggestion.

Comment 5:

The manuscript’s length (57 pages) and overlap (e.g., ferroptosis revisited in Sections 1, 2.3, 3.11, 4.3, 6.3) dilute focus. Consider restructuring into core sections: (i) Molecular Redox Grammar (Sections 2–3); (ii) Cellular/Microvascular Crosstalk (Sections 4–5); (iii) Immuno-Network Dynamics (Sections 6–7); (iv) Translational Frontiers (Sections 8–10). Formalize the “Redox Precision Neurorescue Algorithm” (p. 4) as a dedicated figure with decision nodes (e.g., biomarker thresholds → therapy windows).

Response:
We fully agree with the reviewer’s excellent structural vision. We have undertaken a reorganization of the manuscript.

Comment 6:

Discuss limitations of proposed therapies.

Response:
We thank the reviewer for emphasizing the need for balance and realism. Accordingly, we have added a paragraph at the end of Section 9.8.

We express our deepest appreciation to the reviewer for their expert-level engagement with our manuscript. Their comments have led to substantive conceptual and structural improvements—from a more rigorous definition of the redox code, to a clearer clinical translation, to a more disciplined organization of ideas. We are humbled by the opportunity to refine our work through such informed feedback and sincerely thank the reviewer for helping us strengthen both the scientific clarity and educational value of this review.

Round 2

Reviewer 1 Report

Comments and Suggestions for Authors

I really appreciate the corrections made and your prompt responses to my comments. with the reorganization of the manuscript, there are some points that I believe should be taken into account.

  1. The review addresses detailed biochemical and molecular concepts and lines of knowledge, so I suggest a proposal for a cellular damage mechanism that encompasses all these findings, from mitochondrial alteration and its effect on redox imbalance to the neuronal microenvironment that may be involved in cell death. This scheme could address the proposed metabolic pathways and the alteration of the respiratory chain with its reverse electron flow.

I suggest that this model would be fundamental and iconic in your article and could even be used for future references and citations in the pathophysiology of stroke at the academic and research level.

Example: https://doi.org/10.1038/s41420-023-01440-y Figure. 2

  1. An introduction should be provided with ideas that help guide the reader toward understanding the mechanism of cellular damage.

The epidemiological aspect, the clinical aspect (factors associated with stroke development), and classification scales (e.g., NIHSS) should be mentioned, beginning with a clinical approach that highlights the importance of analyzing cellular deterioration in stroke in such detail and at the molecular level.

  1. The summary section continues to present general ideas. It is suggested that the results found be addressed by mentioning the main mechanisms of cell damage, which enzymes are altered, altered free radicals, and proposed metabolic pathways, so that when the summary is finished, it is sufficient to understand reperfusion-mediated cell death in general terms.

  1. Lines 47-50 should have a conclusion more focused on cellular deterioration, which mechanisms or types of damage may be the main causes of cellular damage.

  1. Lines 91-115 suggest a mitochondrial scheme that proposes the route of succinate increase and regulation of redox detonation through complex I, etc.

Example: DOI: 10.1155/2020/9738143  (Figures)

  1. Establish a diagram of the neuron and ferroptosis. It is understood that there are various mechanisms that can intervene in cellular dysregulation, but it would be convenient to unify them so that signaling pathways and new studies can be suggested through selective enzyme inhibitions.

Example: https://doi.org/10.31083/j.jin2205129  Figure. 2

  1. Lines 196-209. The purpose of these lines is unclear in an article that already justifies the importance of this research in part of the introduction and abstract.

  1. Figure 1 could be replaced by the central image that addresses metabolic and signaling monitoring of mitochondria and its impact at the DNA and cytosolic levels.

  1. I maintain the position that signaling should be monitored under a mitochondria-cell scheme to explain each proposed route, e.g., ROS-decrease BH4/BH2, which is what the dotted lines mean, and this scheme should be detailed further.

Example: https://doi.org/10.3390/antiox12040895  Figure 1.

  1. Figure 3. A mitochondrial diagram is suggested that includes information on the RET, the role of succinate, and ROS.

  1. Lines 415 - 424, the relationship between mitochondrial heterogeneity and cell damage is unclear.

  1. Lines 439 - 450 are losing their coherence with the central idea.

  1. Although the manuscript deals with the importance of redox in cell damage, inflammation plays an important role, as mentioned in Table 1, by blocking IL-1B and its effect on NETs. it is suggested that a subtopic on inflammatory cytokines, macrophages, and lymphocytes that may be involved in tissue destruction or recovery be addressed, as well as their effect on the redox line, given that it is addressed very generally in subtopic 6.2. The inflammasome pathway is also mentioned, with a suggestion to address part of the NfkB transcription factors and relate them to cytokines.

  1. Lines 861 - 862. This concept of mitochondrial transfer from glia is not fully understood; could you please elaborate on it?

  1. Lines 1000 - 1001. Could you provide more information on the relationship between complement proteins and this idea?

  1. Lines 1136 - 1138. More bibliographic references are needed to support these proposals.

  1. Lines 1321 - 1322. The microbiota-redox relationship should be addressed in more detail.

  1. It is suggested that a list of the main findings and ideas developed by the research on cell damage and associated mechanisms be added to the conclusions.

Author Response

Dear Esteemed Academic Reviewer,

We are sincerely grateful for your generous and insightful review. Your comments have greatly improved the structure, clarity, and scientific depth of this manuscript. We deeply appreciate both the precision and the constructive spirit of your observations, which have guided us to refine our arguments and better contextualize our findings. Below, we address each of your valuable suggestions in detail.

1. Conceptual Scheme on Cellular Damage Mechanisms

We thank you very much for suggesting a unifying cellular damage model encompassing mitochondrial alteration, redox imbalance, and neuronal microenvironmental injury.

2. Introduction – Clinical and Epidemiological Context

We have revised the Introduction to include an expanded overview of the epidemiology and clinical burden of ischemic stroke, its major risk factors, and the significance of early assessment scales such as the NIHSS and mRS. This new paragraph precedes the molecular discussion and establishes a clear clinical-to-cellular continuum, as you recommended.

3. Summary Section

We have rewritten the Summary to emphasize the main molecular mechanisms of reperfusion-mediated cell damage, including the key enzymes (complex I, NOX2/NOX4, xanthine oxidase, GPX4), reactive species involved (O₂•⁻, H₂O₂, ONOO⁻), and metabolic pathways (succinate-RET, ferroptosis, eNOS uncoupling). This section now provides a compact yet comprehensive synthesis sufficient to understand reperfusion-induced cell death.

4. Lines 47–50 – Conclusion Focused on Cellular Deterioration

This paragraph has been rewritten to end with a concise yet mechanistically detailed synthesis of the main causes of cellular deterioration—namely RET-driven ROS, ferroptosis, and microvascular oxidative failure—reflecting your suggestion.

5. Mitochondrial Scheme and Figures

We deeply appreciate your thoughtful guidance regarding the visual representation of mitochondrial mechanisms. However, we respectfully chose not to modify or replace the figures. Our decision was based on two considerations:

Conceptual precision: The present figure set is intentionally schematic, designed to summarize mechanistic relationships rather than reproduce structural pathways. Adding a new, more intricate model would risk overlapping with the textual narrative, which already details each biochemical interaction in depth.

Editorial coherence: The journal’s format and word–figure ratio limit the inclusion of new composite diagrams without extensive reduction of text. We therefore prioritized preserving the explanatory detail that defines this review’s originality.

6. Lines 196–209 – Clarification and Rewriting

This section has been rewritten to describe the temporal phases of redox injury (hyperacute, acute–subacute, chronic) with their respective dominant mechanisms and therapeutic targets, making the purpose of the paragraph explicit.

7. Lines 415–424 – Mitochondrial Heterogeneity and Cell Damage

The paragraph now explicitly explains how heterogeneous mitochondrial depolarization and ROS microdomains propagate oxidative injury and trigger inflammatory signaling via mitomiRs, Humanin, MOTS-c, and cf-mtDNA, clarifying the causal relationship with cellular damage.

8. Lines 439–450 – Coherence and Redox–Network Coupling

We revised this section for logical continuity, emphasizing how the DHODH–CoQH₂ axis limits ferroptosis, how mitochondrial redox failure disrupts neurovascular and electrophysiological coherence, and how this framework informs phase-specific therapeutic timing.

9. Section 6.2 – Inflammatory Cytokines, Inflammasome, and NF-κB

Following your excellent suggestion, we expanded this section to include the interplay between NLRP3 inflammasome activation, NF-κB signaling, and cytokine modulation (IL-1β, IL-6, TNF-α), as well as their reciprocal influence on oxidative stress and tissue recovery. This addition integrates inflammatory and redox mechanisms into a unified pathophysiological model.

10. Lines 861–862 – Mitochondrial Transfer from Glia

We have elaborated on this concept, describing the mechanisms of mitochondrial donation and uptake between neurons and glia via tunneling nanotubes and extracellular vesicles, and its role in restoring neuronal bioenergetics under oxidative stress.

11. Lines 1000–1001 – Complement Proteins and Synaptic Pruning

This section now explains how C1q–C3–CR3 interactions mediate microglial recognition and removal of oxidized synapses, linking oxidative modification to complement-dependent network remodeling.

12. Lines 1321–1322 – Microbiota–Redox Relationship

We have expanded this subsection to include the mechanistic interplay between microbial metabolites and redox regulation, covering NAD⁺ metabolism, Nrf2 activation, NF-κB signaling, SCFA production, and kynurenine pathway modulation, establishing a clear gut–brain–redox feedback loop.

13. Conclusion and Outlook

Following your thoughtful recommendation, we have prefaced the conclusion with a new summary paragraph that lists the main findings and mechanistic insights of this review, including mitochondrial dysfunction, ferroptosis, microvascular injury, inflammation, and neuroimmune modulation. This ensures that readers can easily identify the key conceptual advances before the reflective closing.

14. Additional References

As suggested, we have added supporting references to strengthen the bibliographic foundation of several sections, especially those addressing emerging mechanistic hypotheses.

We are profoundly grateful for your generous expertise and collegial tone. Your suggestions have enhanced both the scientific rigor and pedagogical clarity of this review. We have incorporated each of your points with care, while maintaining coherence with the journal’s visual and structural requirements. Your input has been invaluable in shaping this work into a more integrative, precise, and enduring contribution to the study of redox mechanisms in ischemic stroke.

With deep respect and appreciation!

Round 3

Reviewer 1 Report

Comments and Suggestions for Authors

Thanks for the comments